

# Global Marine Gravity Gradient Tensor Inverted from Altimetry-derived Deflection of the Vertical: CUGB2023GRAD

Richard Fiifi Annan[1], Xiaoyun Wan[1*], Ruijie Hao[1], Fei Wang[1]

[1]School of Land Science and Technology, China University of Geosciences (Beijing), Beijing 100083, China

5    *Correspondence to*: Xiaoyun Wan (wanxy@cugb.edu.cn)

**Abstract.** Geodetic applications of altimetry have largely been inversions of gravity anomaly. Literatures wherein Earth's gravity gradient tensor has been studied mostly presented only the vertical gravity gradient. However, there are six unique signals that constitute the gravity gradient tensor. Gravity gradients are signals suitable for detecting short-wavelength topographic and tectonic features. They are derived from double differentiation of the geoid (or disturbing potential); and hence, are susceptible to noise amplification which was exacerbated by low across-track resolution of altimetry data in the past. However, current generation of altimetry observations have improved spatial resolutions, with some better than 5 km. Therefore, this study takes advantage of current high-resolution altimetry datasets to present CUGB2023GRAD, a global (latitudinal limits of ±80º) 1 arc-minute model of Earth's gravity gradient tensor over the oceans using deflection of the vertical as inputs in the wavenumber domain. The results are first assessed via Laplace's equation; whereby the resultant residual gradient is virtually zero everywhere except at high latitudes – icy environments known for contaminating altimetry observations. Due to the absence of similar models from other institutions, the results are further assessed by comparing the vertical tensor component, $T_{zz}$, with equivalent models from Scripps Institution of Oceanography (SIO) and Technical University of Denmark (DTU). The DTU equivalents were derived by multiplying their gravity anomalies by $2\pi k$ in the wavenumber domain. Analysis showed that the inverted $T_{zz}$ averagely deviates from the DTU and SIO equivalents by 0.09 and 0.18 E with corresponding standard deviations of 3.55 and 6.96 E, respectively. Bathymetric coherence analysis of $T_{zz}$ over a section of the western Pacific showed comparable results with the reference models. This study proves that current generation of altimetry geodetic missions can effectively resolve Earth's gravity gradient tensor. The CUGB2023GRAD model data can be freely accessed at https://doi.org/10.5281/zenodo.7710254 (Annan et al., 2023).

## 1 Introduction

It is now 50 years since Skylab, the first satellite altimetry mission, was launched in 1973. This was shortly followed by the GEOS-3 (Geodynamic Experimental Ocean Satellite) and Seasat missions which spanned 1975 – 1979, and 1978, respectively. Satellites that followed these 'first generation' missions have been improvements of knowledge acquired, and technologies developed during the life span of their respective predecessors (Escudier et al., 2018).



Developments in satellite altimetry over the years – such as the improved range accuracy from the Ka-band of Saral/AltiKa

– have resulted in more accurate sea surface heights (SSHs) (Verron et al., 2021, 2018). This, as well as better spatial resolution and other advancements from the Ku-band missions (i.e., the Jason series, HY-2 series, Cryosat-2, Sentinel series, and the recently launched SWOT mission) have enabled diverse applications of satellite altimetry in geodesy, geophysics, glaciology, oceanography and hydrology.

Marine gravity field recovery is the commonest geodetic application of satellite altimetry. Marine gravimetry is important

for submarine navigation (Wan and Yu, 2014), delineating continent–ocean margins (Sandwell et al., 2013), exploring offshore energy resources (Becker et al., 2009), revealing submarine tectonic features buried by sediments (Hwang and Chang, 2014; Sandwell et al., 2014), and deep-sea bathymetry inversion (Annan and Wan, 2020, 2022; Wan et al., 2022a).

For an altimetry satellite's observations to be considered for gravity field recovery, the observations ought to have been acquired during the geodetic mission (GM) phase of the satellite (i.e., in a long repeat orbit). Most satellites begin life in the

exact repeat mission (ERM) phase, where they repetitively observe the same track of ocean surface in a short period; resulting in better temporal resolution at the expense of spatial resolution. The GM phase is considered as end-of-life of the satellite; and it yields higher across-track spatial resolution at the expense of temporal resolution. The higher across-track spatial resolution enables the mapping of short-wavelength features in the gravity field (Andersen et al., 2021). It helps to map out finer details of mean sea surface (MSS), which is used to improve sea level anomalies for the ERM; whereas the GM phase

also benefits from long-term MSS modelled through the ERM phase. The MSS is used to reduce SSH measurements from the GM phase to obtain the geoid – the surface of equilibrium potential. A description of these two satellite phases has been well presented in Andersen et al. (2021).

Although the geoid (or disturbing potential) is the base gravity field signal recovered through satellite altimetry, it is sensitive to long-wavelength features. On the contrary, short-wavelength features, which are of more interest to researchers, are better

revealed through derivatives (i.e., deflection of the vertical, gravity anomaly and gravity gradient tensor) of the disturbing potential. Deflection of the vertical and gravity anomaly are its first derivatives in the horizontal and vertical directions, respectively. Gravity gradients are the second derivatives; and are better at revealing bathymetric and tectonic signatures. Gravity anomalies and gravity gradients can be recovered from geoid heights directly (i.e., through the inverse Stokes formula and double differentiation), or from deflection of the vertical (i.e., through the inverse Vening Meinesz formula and Laplace's

equation). Previous studies have indicated that the use of deflection of the vertical is more accurate, as it minimizes long-wavelength errors (Olgiati et al., 1995; Andersen, 2013).

Even though there are numerous studies about Earth's marine gravity field, most of them are themed on gravity anomaly, and to some degree, on deflection of the vertical. Literatures in which gravity gradients have been studied usually discussed only the vertical component (often denoted as $T_{zz}$) of the gradient tensor although the tensor comprises six unique

components. It suffices to conclude that more research has been conducted on marine gravity anomaly than full tensor of


gravity gradients. Evidently, only Scripps Institution of Oceanography (SIO) releases publicly available gravity gradient models; even those are models of $T_{zz}$ only. One of the reasons for the few literatures on marine gravity gradient tensors is that methods for inverting them from altimetry data are comparatively few, unlike those for inverting gravity anomaly. Another significant justification for this has been the low spatial resolution of altimetry observations in the past. This is

65 because higher differentiation of the disturbing potential results in amplification of high-frequencies, which unfortunately includes noise in the signal (Sideris, 2016; Bouman et al., 2011; Wan et al., 2023). However, current data sets from the GMs of Jason-1, Jason-2, HY-2A, Saral/AltiKa, and Cryosat-2 are more accurate and densified enough to instigate a revisit to altimetry-derived full tensor of gravity gradients. Generally, observations with 8 km across-track spatial resolution are deemed acceptable for gravity field recovery. With the exception Saral/AltiKa, which has variable across-track spatial

resolution (i.e., 1 ~ 15 km) due to its drifting phase (Verron et al., 2021), the spatial resolutions of these other satellites are all better than 8 km (Andersen et al., 2021; Annan and Wan, 2021).

Therefore, this study takes advantage of the abovementioned highly densified data sets to develop CUGB2023GRAD, a global marine gravity gradients product consisting of all six components of the tensor. We compute the gravity gradients in the wavenumber domain through the remove-compute-restore method by using the north-south and east-west components of

75 deflection of the vertical as input signals.

## 2 Data sets

The SSH observations used in this research were computed from the along-track L2P (level-2 plus) sea level anomaly products compiled by AVISO (Archiving, Validation, and Interpretation of Satellite Oceanographic) for each of the aforementioned missions. They can be assessed at ftp://ftp-access.aviso.altimetry.fr/uncross-calibrated/openocean/non-time-

80 critical/l2p/sla. We used all available data sets of Cryosat-2 as of February 2023. For HY-2A and Saral/AltiKa, we used all data sets with cycle numbers $\geq$ 121 and 100, respectively. For Jason 1&2, we used all cycles in the GM phase. The Jason-2 GM observations were derived from geophysical data records accessible at ftp://ftp-access.aviso.altimetry.fr/geophysical-data-record/jason-2. A summarized description of the various satellite missions is presented in Table 1. The equatorial ground tracks of the satellites over a section of the western Pacific (165º E. – 180º E, 5º S – 10º N) are also shown in Fig. 1.

A Gaussian filter, $G_{filter}$, was then applied on the SSHs to minimize noise in the signal. This filtering process also serves as a means of eliminating the time-dependent part of ocean dynamic topography. The filter is given as (Annan and Wan, 2022):

$$G_{filter} = \frac{e^{-\frac{\psi^2}{2\pi^2}}}{\sqrt{2\pi\tau}} \tag{1}$$



where $\tau = 0.2$ is a filter width. $\psi$ is the spherical distance between two consecutive along-track points. If $\left( r, \varphi, \lambda \right)$ is the spherical coordinate of a point (comprising the geocentric radius, spherical latitude and longitude), then for those two consecutive points, $i$ and $j$, the spherical distance, $\psi_{ij}$, between them can be computed through (Sideris, 2016):

$$\cos \psi_{ij} = \sin \varphi_j \sin \varphi_i + \cos \varphi_j \cos \varphi_i \cos \left( \lambda_i - \lambda_j \right) \tag{2}$$

A model of the ocean's mean dynamic topography (MDT) is also needed to reduce the SSH observations to the geoid. Therefore, DTUUH22MDT (Knudsen et al., 2022), an MDT model which incorporated drifter velocity information, was used to remove the time-dependent component of ocean topography. It makes use of the DTU21MSS mean sea surface model that included retracked Cryosat-2 data in coastal and polar areas. DTUUH22MDT was provided by the Technical University of Denmark (DTU, https://ftp.space.dtu.dk/pub/). It is the latest high resolution MDT product developed by DTU.

The remove-compute-restore approach demands the removal of long-wavelengths in the form of an initial gravity signal (i.e., deflection of the vertical). It is later restored in the form of the desired signal (i.e., gravity gradient tensor) after computations. To this end, the global geopotential model EGM2008 (Pavlis et al., 2012) was used to construct the required reference gravity signals. EGM2008 was obtained as spherical harmonic coefficients from the International Centre for Global Earth Models (ICGEM, http://icgem.gfz-potsdam.de/). The reference signals were simulated at maximum degree of 2190 using the GrafLab program developed by Bucha and Janák (2013).

## 3 Methodology

### 3.1 Derivation of Deflection of the Vertical

The altimetry-derived SSH observations are related to the marine geoid, $N$, through the equation,

$$N = SSH - MDT \tag{3}$$

Since deflection of the vertical is the product of first-order differentiation of the geoid in the horizontal direction, it is mostly resolved into its north-south, $\xi$, and east-west, $\eta$, components. The deflection components are computed along the satellite's ground track through (Sideris, 2016):

$$\left. \begin{aligned} \xi &= -\frac{\partial N}{\partial y} \\ \eta &= -\frac{\partial N}{\partial x} \end{aligned} \right\} \tag{4}$$



The deflection components from the different satellites are then gridded using the *surface* module of GMT (Generic Mapping Tools), and are finally fused together based on weights computed using respective error standard deviations relative to deflection components simulated from EGM2008. For the five satellites' signals, $S_1$ $S_2$, $S_3$, $S_4$ and $S_5$, the fused signal, $S$, is obtained as:

$$
\left.
\begin{aligned}
w_1 &= \frac{e_1^2 + e_1 e_2 + e_1 e_3 + e_1 e_4 + e_1 e_5}{\left(e_1 + e_2 + e_3 + e_4 + e_5\right)^2} \\
w_2 &= \frac{e_1 e_2 + e_2^2 + e_2 e_3 + e_1 e_4 + e_2 e_5}{\left(e_1 + e_2 + e_3 + e_4 + e_5\right)^2} \\
w_3 &= \frac{e_1 e_3 + e_2 e_3 + e_3^2 + e_3 e_4 + e_3 e_5}{\left(e_1 + e_2 + e_3 + e_4 + e_5\right)^2} \\
w_4 &= \frac{e_1 e_4 + e_2 e_4 + e_3 e_4 + e_4^2 + e_4 e_5}{\left(e_1 + e_2 + e_3 + e_4 + e_5\right)^2} \\
w_5 &= \frac{e_1 e_5 + e_2 e_5 + e_3 e_5 + e_4 e_5 + e_5^2}{\left(e_1 + e_2 + e_3 + e_4 + e_5\right)^2} \\
w &= w_1 + w_2 + w_3 + w_4 + w_5 = 1 \\
S &= w_1 S_1 + w_2 S_2 + w_3 S_3 + w_4 S_4 + w_5 S_5
\end{aligned}
\right\}
\tag{5}
$$

where $e_1$, $e_2$, $e_3$, $e_4$ and $e_5$ represent inverse error standard deviations.

Instead of assigning a single weight to the values of a satellite 'globally', this study assigns the weights locally. We achieve this by moving a 2°×2° window across the gridded signals of each satellite and EGM2008, and then the weights are assigned per set of 2°×2° grids; thereby localizing the weight assignments. This approach ensures that low accuracy is properly penalized, while high accuracy is encouraged. The locally merged signals in the 2°×2° windows are finally compiled to form a fused global signal.

### 3.2 Derivation of Gravity Gradient Tensor

Marine gravity gradient tensor is derived from the second-order differentiation of the marine geoid in the horizontal and vertical directions. It is a tensor with nine components, of which three are redundant; therefore, there are six unique tensor components. The gravity gradient tensor in the local north-oriented reference frame is defined as (Petrovskaya and Vershkov, 2006; Bouman et al., 2011):



$$
\left.\begin{aligned}
T_{xx} &= \frac{T_{\varphi\varphi}}{r^2} + \frac{T_r}{r} \\
T_{yy} &= \frac{T_{\lambda\lambda}}{r^2 \sin^2 \varphi} + \frac{T_\varphi}{r^2 \tan \varphi} + \frac{T_r}{r} \\
T_{zz} &= T_{rr} \\
T_{xy} &= \frac{T_{\lambda\varphi}}{r^2 \sin \varphi} - \frac{T_\lambda \cos \varphi}{r^2 \sin^2 \varphi} \\
T_{xz} &= \frac{T_\varphi}{r^2} - \frac{T_{\varphi r}}{r} \\
T_{yz} &= \frac{T_\lambda}{r^2 \sin \varphi} - \frac{T_{\lambda r}}{r \sin \varphi}
\end{aligned}\right\}
\tag{6}
$$

$T$ is the disturbing potential, which is related to the geoid, $N$, via the normal gravity, $\gamma$; i.e., $T = \gamma N$. The relationship

between $(\xi, \eta)$ and $T$ can be expressed in the spherical coordinate system as:

$$
\left.\begin{aligned}
\xi &= \frac{1}{\gamma r} \cdot \frac{\partial T}{\partial \varphi} \\
\eta &= -\frac{1}{\gamma r \sin \varphi} \cdot \frac{\partial T}{\partial \lambda}
\end{aligned}\right\}
\tag{7}
$$

Eq. (7) shows that it is possible to deduce the gravity gradient tensor using deflection of the vertical. In order to implement

the remove-compute-restore approach as illustrated in Fig. 2, first compute the residual forms of $\xi$ and $\eta$ as given by Eq.

(8). Another way of arriving at Eq. (8) is to compute the deflection components using residual geoid heights after subtracting

EGM2008-simulated geoid heights from the geoid computed in Eq. (3). This study used the latter approach.

$$
\left.\begin{aligned}
\Delta\xi &= \xi - \xi_0 \\
\Delta\eta &= \eta - \eta_0
\end{aligned}\right\}
\tag{8}
$$

where $\Delta\xi$ and $\Delta\eta$ are the residual north-south and east-west components of deflection of the vertical, respectively. $\xi_0$ and

$\eta_0$ are, respectively, the reference $\xi$ and $\eta$ simulated from EGM2008. From Eqs. (7) and (8), it is obvious to infer that



$$\left.\begin{aligned}
\Delta T_{\varphi} &= \frac{\partial \Delta T}{\partial \varphi} = \Delta \xi \cdot \gamma r \\
\Delta T_{\lambda} &= \frac{\partial \Delta T}{\partial \lambda} = -\Delta \eta \cdot \gamma r \sin \varphi
\end{aligned}\right\} \tag{9}$$

The first derivative of $T$ in the vertical direction produces the radial disturbing gravity gradient, $T_r$. Its residual form, $\Delta T_r$, is computed in the wavenumber domain using the residual components of deflection of the vertical as inputs.

$$\Delta T_r = \mathbf{F}^{-1}\left\{-\frac{i\overline{\gamma}}{k}\left(k_x \mathbf{F}\{\Delta \eta\} + k_y \mathbf{F}\{\Delta \xi\}\right)\right\} \tag{10}$$

where $\overline{\gamma}$ is the mean value of normal gravity. $k = \sqrt{k_x^2 + k_y^2}$ such that $k_x$ and $k_y$ are defined as $\frac{1}{\lambda_x}$ and $\frac{1}{\lambda_y}$, respectively; $\lambda_x$ and $\lambda_y$ are the wavelengths in the horizontal direction. $\mathbf{F}$ and $\mathbf{F}^{-1}$ are the Fourier transform and inverse Fourier transform, respectively.

Having computed the residual signals $\Delta T_{\lambda}$, $\Delta T_{\varphi}$ and $\Delta T_r$ from Eqs. (9) and (10), the derivative property of the Fourier transform is then applied on them to obtain (Wan et al., 2023):

$$\left.\begin{aligned}
\Delta T_{\lambda\lambda} &= \mathbf{F}^{-1}\left\{i2\pi k_y \mathbf{F}\{\Delta T_{\lambda}\}\right\} \\
\Delta T_{\varphi\varphi} &= \mathbf{F}^{-1}\left\{i2\pi k_x \mathbf{F}\{\Delta T_{\varphi}\}\right\} \\
\Delta T_{\lambda\varphi} &= \mathbf{F}^{-1}\left\{i2\pi k_y \mathbf{F}\{\Delta T_{\varphi}\}\right\} \text{ or } \mathbf{F}^{-1}\left\{i2\pi k_x \mathbf{F}\{\Delta T_{\lambda}\}\right\} \\
\Delta T_{\lambda r} &= \mathbf{F}^{-1}\left\{i2\pi k_y \mathbf{F}\{\Delta T_r\}\right\} \\
\Delta T_{\varphi r} &= \mathbf{F}^{-1}\left\{i2\pi k_x \mathbf{F}\{\Delta T_r\}\right\}
\end{aligned}\right\} \tag{11}$$

The components of deflection of the vertical are related to $T_{rr}$ through Laplace's equation, which when expressed in the wavenumber domain results in (Sandwell and Smith, 1997):

$$\Delta T_{rr} = \mathbf{F}^{-1}\left\{-i2\pi\overline{\gamma}\left(k_x \mathbf{F}\{\Delta \eta\} + k_y \mathbf{F}\{\Delta \xi\}\right)\right\} \tag{12}$$

By substituting Eqs. (9) ~ (12) into Eq. (6), residual components of the gravity gradient tensor can now be computed as:



$$
\left.
\begin{aligned}
\Delta T_{xx} &= \frac{\Delta T_{\varphi\varphi}}{r^2} + \frac{\Delta T_r}{r} \\[4pt]
\Delta T_{yy} &= \frac{\Delta T_{\lambda\lambda}}{r^2 \sin^2 \varphi} + \frac{\Delta T_\varphi}{r^2 \tan \varphi} + \frac{\Delta T_r}{r} \\[4pt]
\Delta T_{zz} &= \Delta T_{rr} \\[4pt]
\Delta T_{xy} &= \frac{\Delta T_{\lambda\varphi}}{r^2 \sin \varphi} - \frac{\Delta T_\lambda \cos \varphi}{r^2 \sin^2 \varphi} \\[4pt]
\Delta T_{xz} &= \frac{\Delta T_\varphi}{r^2} - \frac{\Delta T_{\varphi r}}{r} \\[4pt]
\Delta T_{yz} &= \frac{\Delta T_\lambda}{r^2 \sin \varphi} - \frac{\Delta T_{\lambda r}}{r \sin \varphi}
\end{aligned}
\right\}
\tag{13}
$$

Finally, EGM2008-simulated components of the gravity gradient tensor, denoted as: $T_{xx_0}$, $T_{yy_0}$, $T_{zz_0}$, $T_{xy_0}$, $T_{xz_0}$ and $T_{yz_0}$, are then added to the residual tensor components to obtain the gravity gradient tensor.

$$
\left.
\begin{aligned}
T_{xx} &= \Delta T_{xx} + T_{xx_0} \\[4pt]
T_{yy} &= \Delta T_{yy} + T_{yy_0} \\[4pt]
T_{zz} &= \Delta T_{zz} + T_{zz_0} \\[4pt]
T_{xy} &= \Delta T_{xy} + T_{xy_0} \\[4pt]
T_{xz} &= \Delta T_{xz} + T_{xz_0} \\[4pt]
T_{yz} &= \Delta T_{yz} + T_{yz_0}
\end{aligned}
\right\}
\tag{14}
$$

## 4 Results and Analysis

### 4.1 Altimetry-derived Deflection of the Vertical

The altimetry-derived north-south and east-west components of deflection of the vertical (herein referred to as CUGB2023North and CUGB2023East, respectively) are presented in Fig. 3. Using the fused deflection components as dependent variables and the individual components from the satellites as a set of independent variables, we set up an ordinary least squares problem for each deflection component, whereby the solved regression parameters are considered as proxies representing the contribution of each satellite. The least squares problem is repeated for the fused deflection components in which the weights were assigned globally as is usually done in previous studies. For instance, regarding $\xi$, the matrix form of the system of equations is given as:



$$
\begin{bmatrix} \xi_1 \\ \xi_2 \\ \xi_3 \\ \xi_4 \\ \vdots \\ \xi_n \end{bmatrix} = \begin{bmatrix} \xi_{11} & \xi_{12} & \cdots & \xi_{15} \\ \xi_{21} & \xi_{22} & \cdots & \xi_{25} \\ \xi_{31} & \xi_{32} & \cdots & \xi_{35} \\ \xi_{41} & \xi_{42} & \cdots & \xi_{45} \\ \vdots & \vdots & \ddots & \vdots \\ \xi_{n1} & \xi_{n2} & \cdots & \xi_{n5} \end{bmatrix} \cdot \begin{bmatrix} a_1 \\ a_2 \\ a_3 \\ a_4 \\ a_5 \end{bmatrix}
$$
(15)

The design matrix is composed of individual deflection components from the five satellites; thus, a column represents a satellite. The observation vector is composed of values from the fused deflection component. The regression parameters are then used to compute the contributions, $C$, of the satellites.

$$
C_i = \frac{a_i}{\sum\limits_{i=1}^{5} a_i}
$$
(16)

The contributions of the satellites in constructing the deflection of the vertical are summarized in Table 2. In general, the satellites ranked in descending order as: Saral/AltiKa, Cryosat-2, HY-2A, Jason-1 and Jason-2. This ranking is somewhat consistent with Zhu et al. (2020) who had inverted gravity anomaly over the South China Sea. As shown in Table 2, globally assigning weights resulted in the Jason missions collectively accounting for ~40 and ~38 % of $\xi$ and $\eta$, respectively. On the other hand, with the weighting done regionally, they accounted for ~35 % for each of the two deflection components. If

global weights are assigned to the satellites, the contributions of the satellites in $\xi$ is nearly identical at ~20 %. It is known that due to their low orbital inclinations, the Jason missions are better at resolving $\eta$ than other satellites (Annan and Wan, 2021; Wan et al., 2022b). It must be appreciated that the other satellites also resolve $\eta$; however, the signal is noisy due to their high nearly northern inclinations. In both weighting approaches, Cryosat-2 and Saral/AltiKa contributed slightly more than the other missions. Their contributions in resolving $\eta$ are improved because this study minimized the noise by removing

outlying values through histogram analysis and then filtered out remaining noise. Therefore, through effective noise removal, the resolution of $\eta$ by these other satellites can be better than that resolved by the Jason missions. We say so because these two satellites have better spatial resolution than the Jason missions, especially Cryosat-2 (see Fig. 1); and also, Saral/AltiKa has the best range accuracy due to its Ka-band.

Since the deflection components are used to compute the gravity gradients, it is imperative to ensure that their accuracies are

well established. Therefore, they are compared with components of deflection of the vertical developed by the SIO (https://topex.ucsd.edu/pub/global_grav_1min/) and DTU (https://ftp.space.dtu.dk/pub/). The SIO north-south and east-west components used in this analysis are *north_31.1.nc, north_32.1.nc*; and *east_31.1.nc, east_32.1.nc*, respectively. It is worth



noting that DTU does not release models of deflection of the vertical; they instead develop gravity anomaly models directly from geoid heights. Therefore, this study used their gravity anomaly models, DTU17GRA and DTU21GRA, to invert the deflection of the vertical.

In planar approximation, the relationship between residual gravity anomaly, $\delta g$, and $N$ is given as:

$$\delta g = -\gamma \frac{\partial N}{\partial z} \tag{17}$$

Since $(\xi, \eta)$ and $\delta g$ are products of first order derivative of $N$ in the horizontal and vertical directions, respectively, they can be easily inferred from each other in the wavenumber domain. Application of the derivative property of Fourier transform to Eqs. (4) and (17) yields (Sideris, 2016):

$$
\left.
\begin{aligned}
\xi &= \mathbf{F}^{-1}\left\{-i2\pi k_y \mathbf{F}\{N\}\right\} \\
\eta &= \mathbf{F}^{-1}\left\{-i2\pi k_x \mathbf{F}\{N\}\right\} \\
\delta g &= \mathbf{F}^{-1}\left\{2\pi\gamma k \mathbf{F}\{N\}\right\}
\end{aligned}
\right\}
\tag{18}
$$

After substituting and rearranging, $\Delta\xi$ and $\Delta\eta$ can be inverted from $\delta g$ in a remove-compute-restore manner through:

$$
\left.
\begin{aligned}
\Delta\xi &= \mathbf{F}^{-1}\left\{-\frac{ik_y}{\gamma k}\mathbf{F}\{\delta g\}\right\} \\
\Delta\eta &= \mathbf{F}^{-1}\left\{-\frac{ik_x}{\gamma k}\mathbf{F}\{\delta g\}\right\}
\end{aligned}
\right\}
\tag{19}
$$

Subsequently, Eq. (19) was applied on DTU17GRA and DTU21GRA to compute their components of deflection of the vertical; such that, for instance, $\delta g_{DTU17GRA} = \Delta g_{DTU17GRA} - \Delta g_{EGM2008}$. The DTU17GRA and DTU21GRA-derived $\xi$ and $\eta$ are herein referred to as DTU17North, DTU17East; and DTU21North, DTU21East, respectively. Results of the comparative analysis are statistically summarized in Table 3. Since the gravity field signals from both institutions are improvements of previous models, we visually present analysis relative to their latest models in Fig. 4. After removing outlying differences, the magnitudes of the differences between the inverted signals and the reference models generally do not exceed 5 arcsec in both deflection components. The average deviations from the reference models are about 0.10 and 0.03 arcsec in $\xi$ and $\eta$, respectively. Although majority of the differences are zero, as shown by the histograms (Fig. 4b, d, f and h), significant deviations can be seen around tectonic features and islands in the Indian and Pacific oceanic regions. These differences are noticed in Fig. 4a, c, e and g at regions around Malaysia, Indonesia, New Zealand, the Philippines and Japan, to name a few. These observations are as result of the contamination of altimetry observations as the satellites approach



continents and islands. Similarly, the higher absolute differences observed at the polar regions are due the presence of
        icesheets which also degrade altimetry observations. The histograms in Fig. 4 show that the gravity field signals inverted by
        both SIO and DTU are nearly the same

**4.2 Altimetry-derived Gravity Gradient Tensor**

        The inverted gravity gradient tensor is presented in Fig. 5. Gravity gradients are known to be sensitive to topographic
variations; and as such, they are good at revealing short-wavelength bathymetric and tectonic features. Even though some
        tectonic features can be seen in the deflection of the vertical (Fig. 3), they are however better depicted in the various
        components of the gravity gradient tensor. For instance, the outline of the Mid-Atlantic Ridge is well revealed in Fig. 3,
        whereas its spreading is perfectly exposed in addition to its outline in Fig. 5. This observation is an attestation of one key
        characteristic of the gravity potential field: higher differentiations reveal high frequencies. Furthermore, the boundaries of
the African and South-American tectonic plates can be clearly seen in Fig. 5 than in Fig. 3.

        In order to further substantiate the short-wavelength nature of gravity gradients, Fig. 6 presents the GEBCO_2021 bathymetry
        of the same western Pacific region (see Fig. 1) in juxtaposition with the inverted gravity field signals. From Fig. 6, one can
        observe bathymetric signatures in the various gravity field signals, including the two components of deflection of the vertical.
        It is obvious that the bathymetric signatures resolved by the deflection of the vertical have longer wavelengths than those
resolved by the gravity gradients. Additionally, this clearly proves that deflection of the vertical also contain valuable
        bathymetric information that are worth exploiting in the absence of the widely used gravity anomaly and vertical gravity
        gradients (Annan and Wan, 2022).

        To check the accuracy of the gravity gradient tensor, we test the Laplacian equation on the gravity gradient tensor. This can
        be defined as:

$$T_{xx} + T_{yy} + T_{zz} = 0 \qquad (20)$$

        Apart from its ability to tell how accurate the inverted gradient tensor is, the result from the Laplacian equation is also an
        indication of the effectiveness of the inversion method used to derive the signals. The residual gradient signal shown in Fig.
        7 is the result of the Laplacian operation. It can be seen that the residuals are practically zero everywhere except at higher
        latitudes (typically beyond latitudes ±60°) where the altimetry observations at the polar regions are known to be contaminated
by the icy environments. The average residual gradient is -0.0024 E, with a standard deviation of 0.7287 E. The high accuracy
        reported in Fig. 7 is an alternative interpretation of the accuracy of the altimetry observations. Also, it consequently serves
        as an indicator of the accuracy of the deflection of the vertical. This is because each component of the gravity gradient tensor
        is computed from the same north-south and east-west components of the deflection of the vertical.





Moreover, since there are no publicly known models of gravity gradient tensors for comparison, we validated the accuracy

of the inverted gravity gradient tensor by comparing its vertical component, $T_{zz}$, with equivalent models from SIO and DTU. The SIO vertical gravity gradient (VGG) models used for this analysis are *curv31.1.nc* and *curv32.1.nc*. Again, since DTU does not release gravity gradient models, we constructed DTU17VGG and DTU21VGG in the remove-compute-restore manner by simply multiplying residual forms of DTU17GRA and DTU21GRA by $2\pi k$ in the wavenumber domain, respectively. After this, EGM2008-simulated VGG is restored to get DTU17VGG and DTU21VGG. The results from the

latest reference models (i.e., SIO32.1VGG and DTU21VGG) are visualized in the map views and histograms of Fig. 8. After eliminating differences that exceed three times the standard deviation from the mean difference, the mean deviations from the reference VGGs are all less than 0.20 E. These are shown in the statistical summary presented in Table 4. It can be seen from Fig. 8 and Table 4 that the inverted $T_{zz}$ is closer to the signals from DTU than those from SIO.

Additionally, the coherency between the inverted $T_{zz}$ and GEBCO_2022 bathymetry of the western Pacific region was

computed. This is juxtaposed with corresponding coherencies derived from the VGGs obtained from SIO and DTU in Fig. 9. The curves in Fig. 9 are nearly identical, with the main differences seen at the high and low wavelengths. The small coherency values at the low and high wavelengths are caused by upward continuation of gravity field from the seafloor to the sea surface, as well as isostatic compensation due to submarine topography (Smith and Sandwell, 1994). Analysis of results shows that with a minimum coherency of 0.5, the inverted $T_{zz}$ can detect bathymetric features within a wavelength

band of 20 – 345 km. The VGGs from SIO can detect features within 13 – 335 km; whereas DTU17VGG and DTU21VGG can reveal bathymetric features within wavelength bands 20 – 348 and 15 – 345 km, respectively. Bathymetric features with wavelengths within 25 – 230 km would be detected with higher accuracy than features outside this range. This is because the coherencies of these wavelengths are greater than or equal to 0.70 in each of the five vertical gravity gradients.

In summary, the gravity field signals developed in this study are solely from satellite altimetry observations; whereas the

reference models from SIO and DTU are improvements of previous models which incorporated data from different sources including shipboard gravimetry. The gravity gradients presented in this paper proves that the high spatial density and SSH accuracy of currently available GM datasets are capable of resolving the various components of Earth's gravity gradient tensor over the oceans. The results from this study further substantiates a statement by Sandwell et al. (2013) who had recently asserted that gravity field signals inverted from current generation of altimetry datasets are becoming more superior in quality

than most of the publicly available shipborne gravimetry datasets. Therefore, if the geoscience community would invest similar efforts in the techniques of inverting gravity gradients like has been invested in gravity anomaly, the accuracy of future models of gravity gradient tensor would be improved. We say this in light of the high range accuracy from the Ka-band mission (i.e., Saral/AltiKa), as well the high across-track sampling from Cryosat-2 and the recently launched SWOT mission which incorporates interferometric technology.



### 4.3 Data availability

The global marine gravity gradient tensor model, CUGB2023GRAD, is available at the ZENODO repository, https://doi.org/10.5281/zenodo.7710254 (Annan et al., 2023). The dataset consists of GMT-readable geospatial grids in NetCDF file format (i.e., vector of latitudes, vector of longitudes, and matrix of gravity gradients).

## 5 Conclusion

Components of deflection of the vertical have been inverted from altimetry-derived SSHs; and used as input signals to invert marine gravity gradient tensor over the globe. Analysis of results indicate that the contributions of the satellites in resolving the north-south deflection component are almost equal at ~20 % each. The Jason missions are good at resolving the east-west component of deflection of the vertical due to their relatively low orbital inclination of 66°. However, with effective outlier removal and filtering, and localized weight assignment, Cryosat-2's high spatial resolution and Saral/AltiKa's high range accuracy can enable them to contribute more east-west deflection components than each of the two Jason missions. The HY-2A mission ranked slightly ahead of the Jason missions. The resultant gravity gradient tensor was assessed via the Laplacian equation; with the corresponding residual gradient having magnitudes close to zero across the globe, except at latitudes exceeding ±60°. These are regions dominated by icesheets, and are known to degrade altimetry observations. Comparison of the inverted $T_{zz}$ with equivalent signals from SIO and DTU showed that it averagely deviates from the SIO and DTU equivalents by 0.18 and 0.09 E, respectively. Further analysis of the inverted $T_{zz}$ through bathymetric coherence analysis showed that it compares well with the reference models from SIO and DTU. The average across-track sampling of current generation of altimetry observations is better than the 8 km minimum required for gravity field inversion. Therefore, with the anticipated higher accuracy and better spatial resolution from the recently launched SWOT mission and upcoming Ka-band altimetry missions, coupled with an increase in research interest and investment, the accuracy of future gravity gradient tensor models would be improved.

### Author contribution

RFA and XW conceived the idea for this paper and inverted the gravity field signals. All authors contributed to the various analyses. XW secured funding and supervised the work. RFA prepared the original manuscript; all authors contributed to review and editing.

### Competing interests

None of the authors has any competing interests.





**Acknowledgements**

The authors are grateful to DTU and SIO for making their respective gravity field signals accessible to us. The Generic Mapping Tools (Wessel et al., 2019) was used for drawing the maps and for performing some of the analyses. AVISO is highly appreciated for providing the altimetry datasets. The ICGEM and British Oceanographic Data Centre are also appreciated for making EGM2008 and GEBCO_2022 available. We are very grateful to Blažej Bucha for providing us the GrafLab program.

**Financial support**

This research was funded by the National Natural Science Foundation of China (Nos. 42074017, 41674026).

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



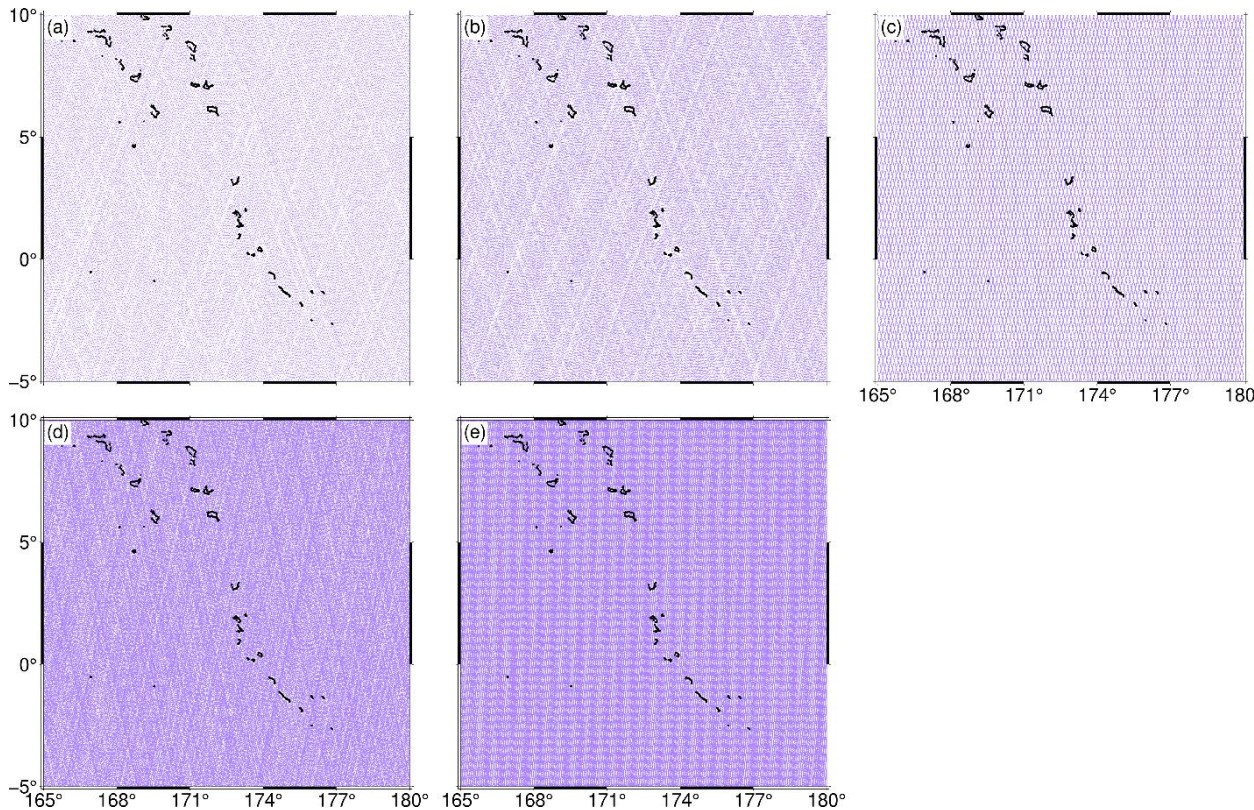

Figure 1. Equatorial ground tracks of (a) Jason-1, (b) Jason-2, (c) HY-2A, (d) Saral/AltiKa and (e) Cryosat-2





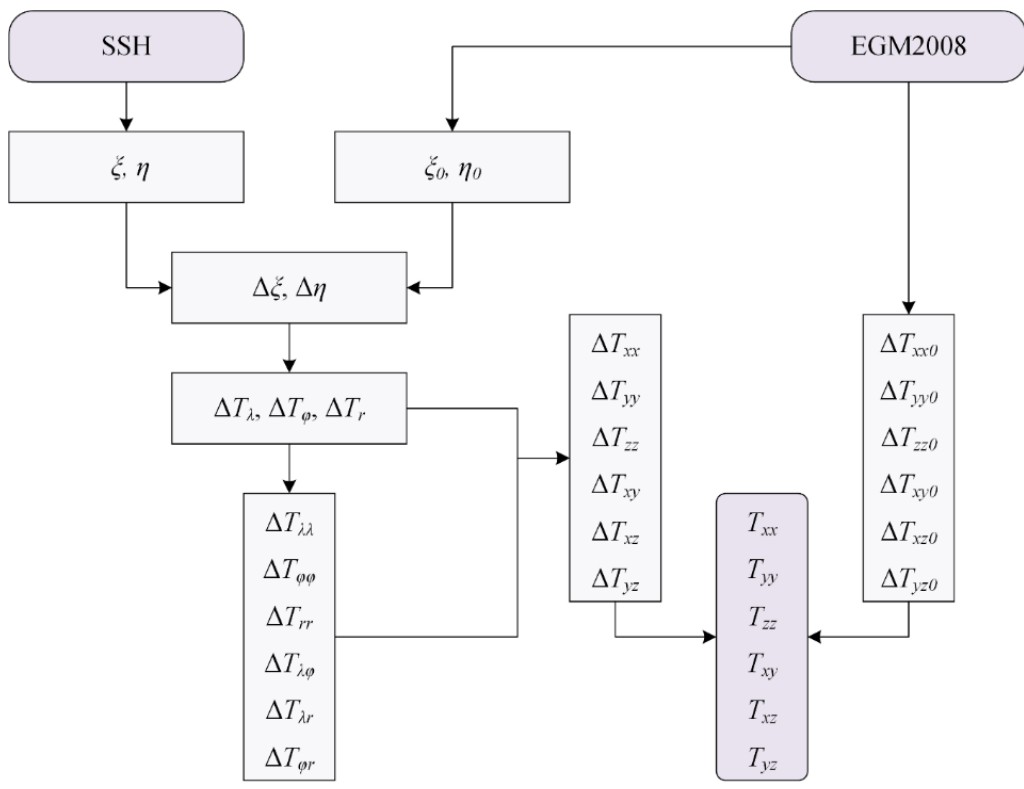

Figure 2. Illustration of the remove-compute-restore approach used

Figure 3. Altimetry-derived deflection of the vertical: (a) CUGB2023North, and (b) CUGB2023East





Figure 4. Assessment of inverted deflection of the vertical: (a) map view and (b) histogram of deviation of CUGB2023North from SIO31.1North; (c) map view and (d) histogram of deviation of CUGB2023East from SIO31.1East; (e) map view and (f) histogram of deviation of CUGB2023North from DTU21North; (g) map view and (h) histogram of deviation of CUGB2023East from DTU21East

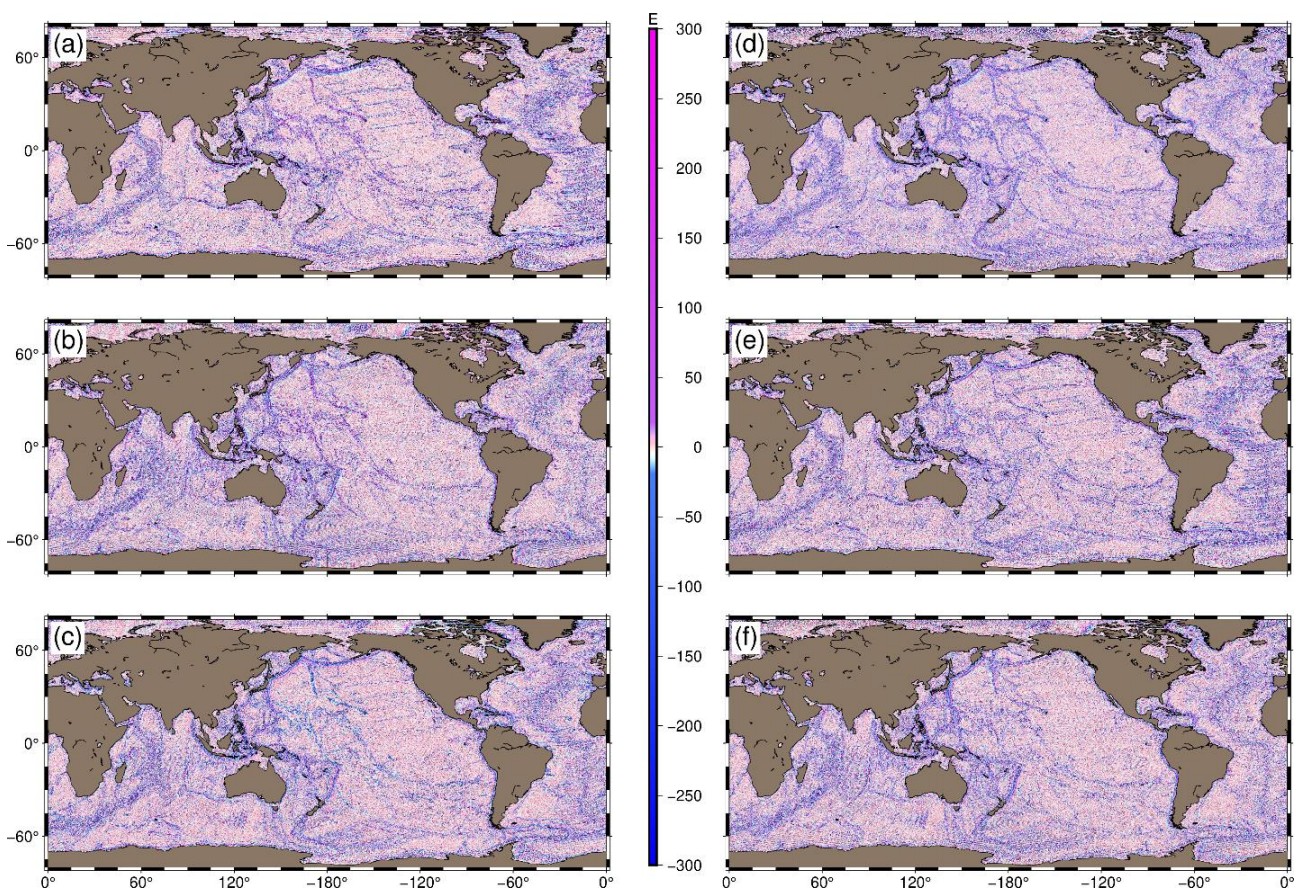

Figure 5. Altimetry-derived gravity gradient tensor: (a) $T_{xx}$, (b) $T_{yy}$, (c) $T_{zz}$, (d) $T_{xy}$, (e) $T_{xz}$, and (f)-$T_{yz}$

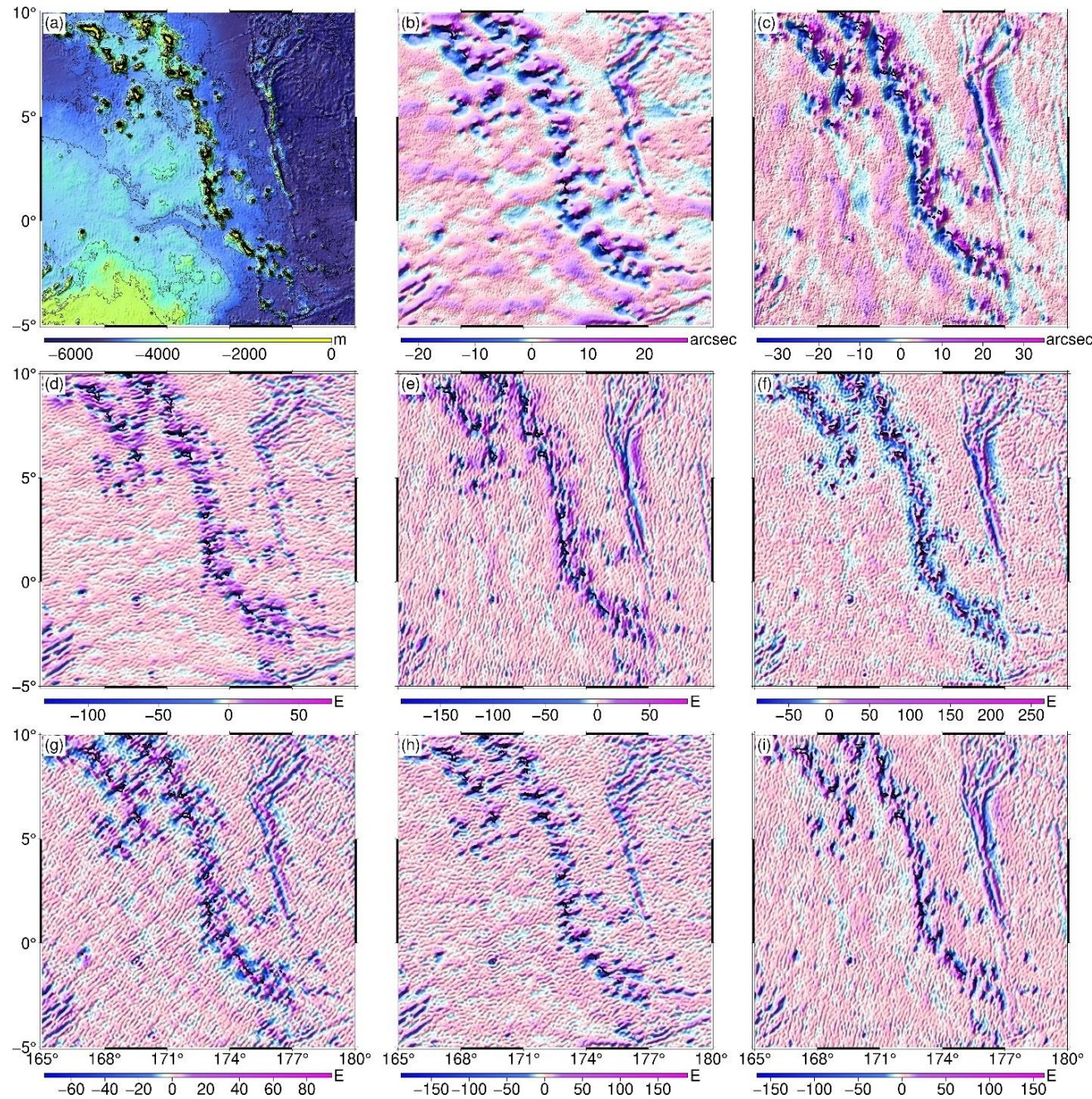

Figure 6. Bathymetry and inverted gravity field signals of western Pacific: (a) seafloor topography, (b) $\xi$, (c) $\eta$, (d) $T_{xx}$, (e) $T_{yy}$, (f) $T_{zz}$, (g) $T_{xy}$, (h) $T_{xz}$, and (i) $T_{yz}$



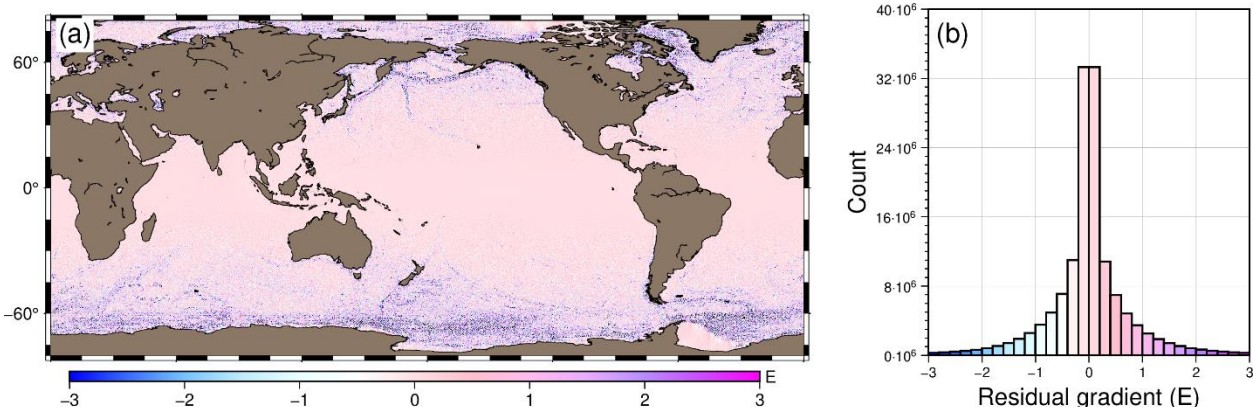

Figure 7. Result of the Laplacian operation: (a) map view and (b) histogram of residual gradient signal

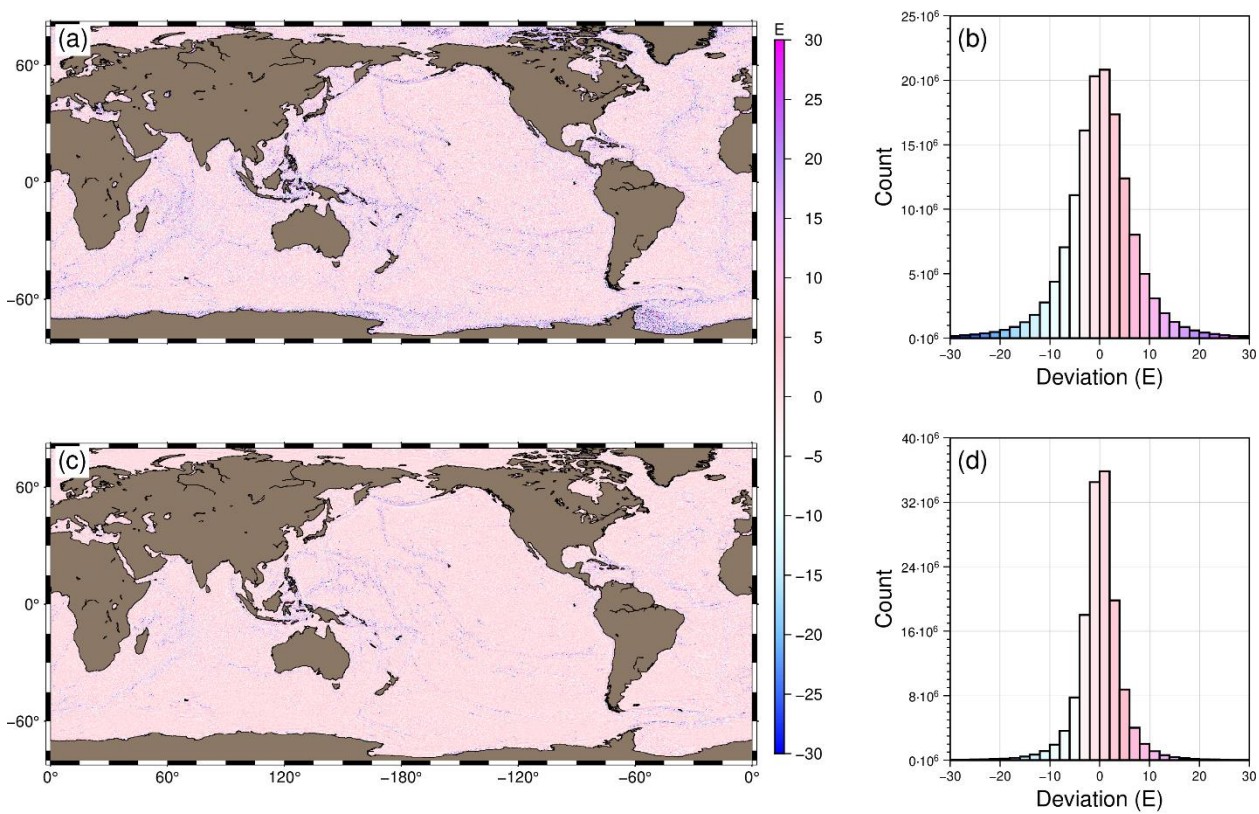

Figure 8. Assessment of inverted gravity gradients: (a) map view and (b) histogram of deviations of inverted $T_{zz}$ relative to

SIO31.1VGG; and (c) map view, and (d) histogram of deviations of inverted $T_{zz}$ relative to DTU21VGG





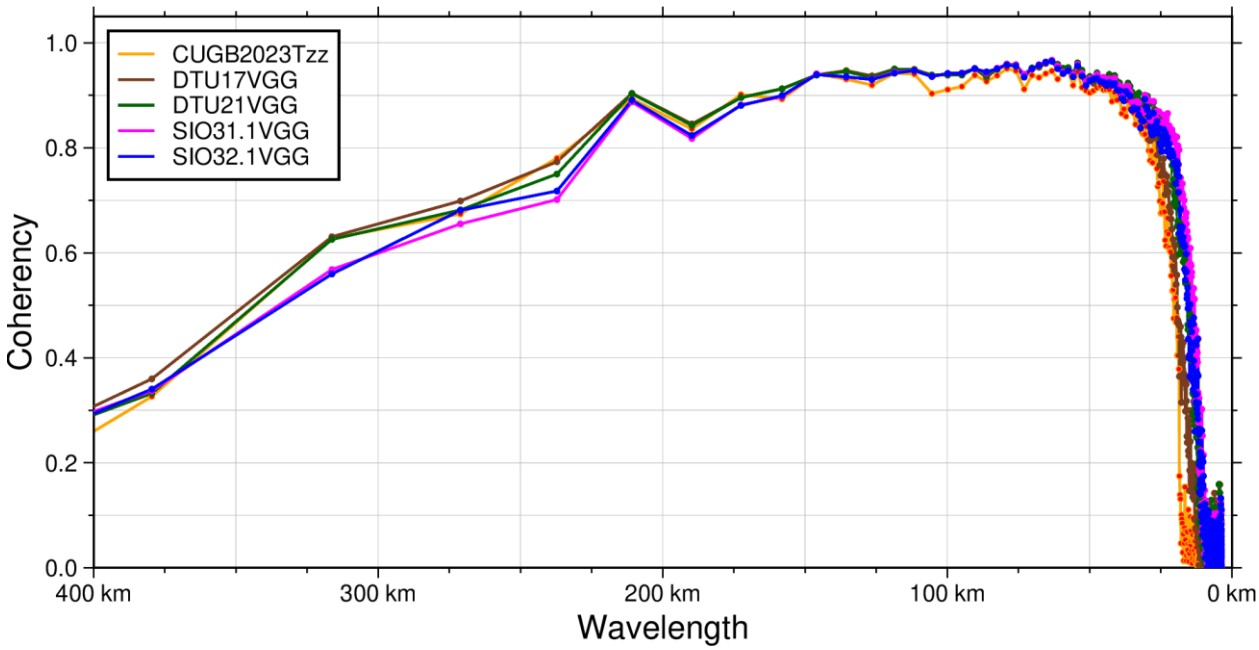

Figure 9. Coherency between vertical gravity gradient and bathymetry of the western Pacific region

Table 1. Summary description of the satellite missions

| Satellite | Cycles used | Equatorial spatial resolution (km) | Temporal resolution (days) | Latitudinal extent (º) |
|---|---|---|---|---|
| Jason-1 | 500 – 537 | ~7.5 | 406 | ±66 |
| Jason-2 | 500 – 537 & 600 – 644 | ~7 | 371 | ±66 |
| HY-2A | 121 – 288 | ~8.7 | 168 | ±81 |
| Saral/AltiKa | 100 – 166 | ~3.2 | – | ±81.5 |
| Cryosat-2 | 007 – 130 | ~2.5 | 369 | ±88 |

Table 2. Contribution of each satellite in resolving the components of deflection of the vertical

| Satellite | Regional assignment of weights | | Global assignment of weights | |
|---|---|---|---|---|
| | Contribution in $\xi$ (%) | Contribution in $\eta$ (%) | Contribution in $\xi$ (%) | Contribution in $\eta$ (%) |
| Jason-1 | 16.77 | 18.87 | 19.81 | 20.54 |
| Jason-2 | 18.12 | 15.92 | 20.23 | 17.73 |
| HY-2A | 19.50 | 20.72 | 19.36 | 19.40 |
| Saral/AltiKa | 22.70 | 22.81 | 20.29 | 21.30 |
| Cryosat-2 | 22.91 | 21.68 | 20.30 | 21.04 |





Table 3. Assessment of inverted deflection components relative to reference models (unit: arcsec)

| Difference | Minimum | Maximum | Mean | Standard deviation |
|---|---|---|---|---|
| CUGB2023North – DTU17North | -3.9000 | 4.0724 | 0.0922 | 1.0819 |
| CUGB2023North – DTU21North | -4.0102 | 4.1796 | 0.0926 | 1.1092 |
| CUGB2023North – SIO31.1North | -4.2612 | 4.4447 | 0.1096 | 1.1604 |
| CUGB2023North – SIO32.1North | -4.2200 | 4.4005 | 0.1081 | 1.1560 |
| DTU17North – SIO31.1North | -1.3910 | 1.4128 | 0.0157 | 0.2525 |
| DTU17North – SIO32.1North | -1.2474 | 1.2675 | 0.0135 | 0.2296 |
| DTU21North – SIO31.1North | -1.2648 | 1,2885 | 0.0165 | 0.2385 |
| DTU21North – SIO32.1North | -1.2444 | 1.2666 | 0.0148 | 0.2398 |
| DTU17North – DTU21North | -1.0461 | 1.0456 | -0.0011 | 0.2356 |
| SIO31.1North – SIO32.1North | -0.7391 | 0.7391 | -0.0014 | 0.1312 |
|  |  |  |  |  |
| CUGB2023East – DTU17East | -3.3738 | 3.4047 | 0.0301 | 0.8484 |
| CUGB2023East – DTU21East | -3.4677 | 3.4961 | 0.0301 | 0.8749 |
| CUGB2023East – SIO31.1East | -3.7362 | 3.7697 | 0.0323 | 0.9465 |
| CUGB2023East – SIO32.1East | -3.6828 | 3.7205 | 0.0352 | 0.9387 |
| DTU17East – SIO31.1East | -1.5897 | 1.5927 | 0.0016 | 0.3311 |
| DTU17East – SIO32.1East | -1.4491 | 1.4561 | 0.0041 | 0.3060 |
| DTU21East – SIO31.1East | -1.4424 | 1.4457 | 0.0022 | 0.2854 |
| DTU21East – SIO32.1East | -1.3561 | 1.3633 | 0.0047 | 0.2760 |
| DTU17East – DTU21East | -0.8415 | 0.8405 | -0.0004 | 0.2209 |
| SIO31.1East – SIO32.1East | -0.9391 | 0.9391 | 0.0024 | 0.1850 |

Table 4. Assessment of CUGB2023 $T_{zz}$ relative to reference models (unit: E)

| Difference | Minimum | Maximum | Mean | Standard deviation |
|---|---|---|---|---|
| CUGB2023$T_{zz}$ – DTU17VGG | -12.2590 | 12.3931 | 0.0811 | 3.3533 |
| CUGB2023$T_{zz}$ – DTU21VGG | -13.8652 | 14.0605 | 0.1098 | 3.7422 |
| CUGB2023$T_{zz}$ – SIO31.1VGG | -28.3698 | 28.8155 | 0.1862 | 7.0488 |
| CUGB2023$T_{zz}$ – SIO32.1VGG | -27.5944 | 28.0277 | 0.1772 | 6.8686 |
| DTU17VGG – SIO31.1VGG | -23.9927 | 24.0958 | 0.0453 | 5.2901 |
| DTU17VGG – SIO32.1VGG | -23.1077 | 23.2134 | 0.0499 | 5.1188 |
| DTU21VGG – SIO31.1VGG | -23.3505 | 23.4480 | 0.0396 | 4.9822 |
| DTU21VGG – SIO32.1VGG | -22.4992 | 22.5987 | 0.0426 | 4.8124 |
| DTU17VGG – DTU21VGG | -7.8684 | 7.8704 | 0.0018 | 1.6527 |
| SIO31.1VGG – SIO32.1VGG | -15.0662 | 15.0662 | 0.0034 | 2.7336 |