# Peer review of "Global Marine Gravity Gradient Tensor Inverted from Altimetryderived Deflections of the Vertical: CUGB2023GRAD"

_Earth System Science Data, 2023_

## Referee Comment (RC1)

Review of essd-2023-85, "Global marine gravity gradient tensor inverted from altimetry-derived deflection of the vertical: CUGB2023GRAD", by Walter H. F. Smith (ORCID, ResearcherID)

General Comments

In my opinion, this manuscript does not meet the standard of enabling the user to evaluate whether the data product CUGB2023GRAD will be useful, whether it is an improvement on existing products, or even whether or how the authors have taken steps to avoid circular reasoning. Critical issues in data editing, filtering, and computation are not adequately described. I also believe that the input data sets may not be the best ones for this kind of analysis.

The work described seems to rely heavily on the previous works of the Danish group led by Ole Andersen and the Scripps group led by David Sandwell. While a few papers from these groups are cited, often the most relevant ones are not cited. It would be helpful to have reviews from Drs. Andersen and Sandwell.

Specific Comments

The strong coherence of the data with the GEBCO bathymetry (manuscript Figure 9 and its discussion) should not be taken as a measure of the quality or value of CUGB2023GRAD, because depths predicted from satellite altimetry are in the GEBCO product. In order that the reasoning not be circular here, one would need to demonstrate that the coherence had been computed from a portion of the GEBCO product that contained almost entirely in situ measured depths, and not depths estimated from altimetry. The GEBCO Source ID grid could be a help here.

As the manuscript makes clear, all six gradient tensor elements arise from differentiation of one scalar quantity (the disturbing potential), and so these elements are six different views or characterizations of one set of information. The manuscript does not demonstrate the utility of having all these different views of the same information.

The manuscript presents the trace (sum of the diagonal elements) of the gradient tensor and suggests that the quantitative value of this sum is an evaluation of the product. But since the equations used all derive from the assumption that the gravity field obeys Laplace's equation, the trace ought to be zero by definition. The manuscript does not present any way for the reader to

undestand quantitatively the significance of a non-zero trace: how does it compare to the noise in the data, noise in the model components, limitations on the resolution of each quantity, etc.?

The manuscript has too many equations presenting the general theory, as this could have been summarized with a citation to a standard textbook. Some of these equations are given in spherical coordinates and some in Cartesian coordinates, and it is not clear which coordinates are used for the calculations done to produce CUGB2023GRAD. What the manuscript needs to do is to explain how the calculations in the 2 degree by 2 degree patches of Earth surface area were carried out. I presume they used Fourier transforms on Cartesian coordinates after a remove-restore procedure.

In my opinion 10.1016/j.asr.2019.09.011 does a very good job of demonstrating quantitatively the contribution of each satellite altimeter mission to the overall marine gravity field model, treating the east-west and north-south components separately and treating each as functions of latitude, and showing what weight should be given to each. The present manuscript does not do this very well. Perhaps the present study's analysis of what weight to give each satellite mission in deriving each of the tensor quantities in each of the 2x2 squares might furnish some interesting information, but that information is not presented here.

The filtering of the data is an important detail, but the equation describing the filter (Equation 1) is wrong: if tau is the filter width parameter then tau-squared should appear somewhere in the argument to the exponential in Equation 1. Another minor point: I assume that the data to be filtered are very closely spaced, and in that case computing the filter using Equation 2 followed by the inverse cosine will be quite inaccurate.

The input data used are available only at the "1 Hz" nominal sampling rate, for many of the altimeters included in this study. As many papers by Sandwell and his colleagues for over 30 years have shown, "1 Hz" data are down-sampled from boxcar averages of the original data, which have nominal sampling rates of 10, 20 or 40 Hz, depending on the satellite; the boxcar averaging has bad side-lobes; and the 1-Hz downsampling aliases sidelobe energy into long along-track wavelengths, spoiling the accuracy of the resulting along-track deflections of the vertical. For this reason, Sandwell and colleagues have taken great pains to design specialized filters and downsampling rates. Therefore I believe that the accuracy and utility of CUGB2023GRAD may be limited by the fact that it starts from "1 Hz" data. (The along-track filter design description in 10.1029/95JB01308 pre-dates the development of two-pass retracking [10.1093/gji/ggt469] and so the filters now used have different pass- and stop-band specification than what is described in 95JB01308.)

An important detail is the removal of the non-geoidal signals from the sea surface height. One of these, the Mean Dynamic Topography, is mentioned in Equation 3. But others (tides, transient dynamic signals, etc., as well as errors in radar path delays, sea state bias, etc.) are not mentioned. One wonders how this was done. It will have an important impact on the quality of CUGB2023GRAD.

Equation (4) correctly shows that the north and east components of deflection are the coresponding partial derivatives of the geoid height anomaly, but in the Introduction these deflection components are incorrectly described as derivatives of the disturbing potential. The correct relationship requires relating the geoid height to the disturbing potential, such as via Bruns formula.

It is not correct to say there has been little prior work on the vertical gravity gradient; in addition to 10.1126/science.1258213 , 10.1029/2020JB020017 should also be cited.

The Generic Mapping Tools should be cited for the program *surface* at line 111; which citation depends on which version was used: https://www.generic-mapping-tools.org/cite/

If the authors, or anyone reading this review, needs to download PDF reprints of David Sandwell's papers, they are available for free download (no paywall) here.

---

## Referee Comment (RC2)

The altimetric data is widely used to explore the oceanic gravity field, but most on gravity anomaly. This paper presented a method to construct marine gravity gradient tensors from altimetric deflections of the vertical, and given a global model CUGB2023GRAD.

Some revisions are required:

(1) All over the paper, "deflection of the vertical" should be "deflections of the vertical", including both north-south and west-east components.

(2) Line 9, in the abstract, "They are derived from double differentiation of the geoid (or disturbing potential)". Its not clear. Gravity gradients are the second second derivative of disturbing potential. The geoid should be connected with disturbing potential by Bruns formula.

(3) Line 37, altimetric gravity anomaly is widely used to predict bathymetry, too few references are given. Many paper from Smith, Sandwell and Anderson should be referenced.

(4) Line51-52, "Deflection of the vertical and gravity anomaly are its first derivatives in the horizontal and vertical directions, respectively. ", Again, you should make it clear, geoid and disturbing potential are different.

(5) Line 130,the first formula in eq.(7), there should be a "-".

(6) In order to validate the gravity gradients results, the coherence between CUGB2023GRAD and GEBCO_2021 were computed. In order to avoid circular validation problems, I recommend adopting multi-beam bathymetry grid from JAMSTEC or NCEI to replace GEBCO_2021 model.

---

## Referee Comment (RC3)

**Review of Global Marine Gravity Gradient Tensor Inverted from Altimetry derived Deflection of the Vertical: CUGB2023GRAD**

The authors computed components of gravity gradient tensor using altimetry derived deflections of vertical. They combined different altimetry satellite data assigning weights to them. Applying remove-compute-restore technique they developed CUGB2023GRAD grid which is publicly available.

The paper does not explain the strength of the method applied nor the potential benefit of their product for use which I think is very important. On the one hand, the methodology needs to be expanded in general. But on the other hand, there are various equations which I think is very lengthy. Some adjustment is needed therefore in balancing the content. I am not sure if an Appendix can be added.

I can imagine readers being interested in learning more about the benefits and applications of the outcome of this work. The authors could be more convincing and provide some details also w.r.t. the literature. This is not provided in the current version. I also think that some of the figures can be explained better.

The use of EGM2008 needs to be justified in general. For marine related gravity field, I can imagine GOCE data also should be included in the GGM used. I wonder whether GEBCO grid is the best option to compare with. Would there be any case one compares the outcome of this work for instance w.r.t shipborne measurements directly, maybe along track measurements of bathymetry?

Some detailed comments:

Line 101: Please use the ESSD reference for the ICGEM service.

Line 111: Please explain why the weights are calculated based on EGM2008? What would be an alternative to this?

Line 133: Preference for using EGM2008 needs to be justified.

Last sentence on Page 6 is confusing.

Line 171: Why somewhat? Was it unexpected?

Again, some sections are unnecessarily lengthy and some sections are not explained as much needed.

Line 193: No parenthesis for deflections of vertical

Line 301: GEBCO_2021 or 2022? I believe the two are used in the paper. Please explain why? Would there be any in-situ data available for this purpose? Would it be more reliable to do comparisons w.r.t shipborne or other in-situ measurements?

Eq 1: Not pi but tao in the dividend. Please double check the equation.

Some well-known formulations have been repeated. I think this is not needed.

Please explain why you have larger differences in the East direction, please explain the differences between Figures 4c-d and 4g-h

Please explain the coordinate systems used when needed in the text and in which the Txx and the other 5 are given.

Figure 8b: Differences w.r.t SIO31.1VGG look larger in the map. Are they consistent with the histogram? Please double check.

Could you please explain the outcome and benefit of the representation of Fig. 9?

Please use the GMT reference when needed.

---

## Community Comment (CC2)

Line 6: Here what are literatures?
**Response:** 'literatures' in this context means previous studies. We will change it to 'previous studies'.

Line 41: Some altimetry satellite missions consider their GM phase in their start or middle life of satellites, not all end of life.
**Response:** Thank you for this comment. Your point is very true. We will rephrase the sentence as: The GM phase is usually considered as end-of-life of the satellite though some altimetry missions consider their GM phase in their start or middle life of the satellite.

Line 49: Here should be geoid model instead of geoid.
**Response:** We will change it as you have mentioned. Thanks very much.

Line 58: Here what are literatures?
**Response:** 'literatures' in this context means previous studies. We will change it to 'previous studies'.

Section 2: Only GM data of 5 satellite missions are used in the study. Why not use the other satellite altimeter data include ERM data? What errors are corrected in SSHs? How to assess the precision of SSHs? How to distinguish SSHs' frequencies?
**Response:** It is true that ERM datasets are also useful. However, as already mentioned in the manuscript, gravity field recovery relies more significantly on high spatial resolution data, which is available in GM datasets rather than in ERM datasets. Typical example is the work of Sandwell et al. (2019), the research paper for SIO's gravity field models. Their study entirely used GM datasets from Geosat, ERS-1, Jason-1/2, CryoSat-2 and Saral/AltiKa. They concluded that "originally, Geosat and ERS-1 were the most important altimeters for recovery of the marine gravity field". This means that the contributions from Jason-1/2, CryoSat-2 and Saral/AltiKa supersedes those from Geosat and ERS-1. In fact, their results showed that ERS-1 no longer provides any significant improvement. The implication here is that, if the GM datasets of Geosat and ERS-1 contributed minimally, then we should not expect better from their ERM versions. Therefore, the conclusions from Sandwell et al. (2019) are the main reasons for our use of these 5 satellites.

The errors corrected in SSHs are defined in Section 3.2.3.2 of Along-track Level-2+ (L2P) SLA Product Handbook. When we revise the manuscript, we will include a sentence to that effect in the manuscript. In this study, the precision of the SSHs is viewed from the precision of the DOVs (Table 3). This is because the DOVs are derived directly from the SSHs. In other words, accurate DOV components implies that the SSHs are also accurate. The frequency of the SSHs is 1 Hz as stated in the Along-track Level-2+ (L2P) SLA Product Handbook.

Sandwell DT, Harper H, Tozer B, Smith WHF (2019) Gravity field recovery from geodetic altimeter missions. Adv Space Res. https://doi.org/10.1016/j.asr.2019.09.011

Line 93: What earth gravity field model is used in the MDT model? Not egm08??
**Response:** According to Knudsen et al. (2022), the earth gravity field model used in the MDT model is XGM2019e.

Knudsen, P., Andersen, O., Maximenko, N., and Hafner, J.: A new combined mean dynamic topography model – DTUUH22MDT, ESA Living Planet Symposium 2022, Bonn, Germany, 2022.

Line 110: What are x and y?
**Response:** *x* and y refer to the longitude and latitude, respectively. We will modify when revising the manuscript. Thanks very much for drawing this to our attention.

Line 112: How to precisely determine weight?
**Response:** Firstly, deviations of components of DOV for each satellite are determined relative to EGM2008-derived components of DOV. These deviations are used to compute error standard deviation (which is inversed to get $e$) for each satellite for each DOV component. Assuming we want to compute the weights needed to fuse $\xi$, we compute $e = 1/\operatorname{std}(\xi_{satellite} - \xi_{EGM2008})$ for a satellite. This is repeated for the remaining four satellites. The calculated $e$ values are then inputted into Eq. 5 to compute the weights needed to arrive at the fused $\xi$ signal.

Note that due to the 66⁰ inclination of the Jason missions, the weight assignment in Eq. 5, and consequently satellite contribution analysis, were conducted within latitudinal bounds of ±60⁰. Individual satellite DOV components outside of this latitudinal range were fused using data from HY-2A, Saral/AltiKa and Cryosat-2 only. We will modify the manuscript accordingly. Thank you very much for this comment.

Line 119: Why to use 2 degrees?
**Response:** We used 2 degrees because it provided a good trade-off for computer memory consumption. A smaller value would significantly increase the number of smaller grids which occupied a huge proportion of our computer's memory capacity; whereas a bigger value would also incorporate low accuracy points farther away.

Line 125: How to define the local reference frame in detail? One local reference frame is only used in one small area.
**Response:** Thank you for this comment. You are right, the reference frame is used in one small local area. The local reference frame is north-oriented (i.e., *x, y, z*); with *x* referring to the direction along latitude to the north, *y* referring to the direction along longitude to the west, and *z* referring to radial direction to outside of the Earth.

Line 128: Here normal gravity should be normal gravity in geoid.
**Response:** We will change it. Thank you very much.

Line 143: How to calculate λ? How about λ=0?
**Response:** Thanks for the comment. When λ=0, we set the value to be very small, such as 1e-6. With $[N, M] = size\left(\text{Gravity field signal}\right)$, and $(\Delta x, \Delta y)$ being the grid spacings along $(x, y)$ axes, $(k_x, k_y)$ are defined as:

When *M* or *N* is even,

$$k_x = \frac{1}{M \cdot \Delta x}\left(\frac{-M}{2}, \cdots, 0 \cdots, \left(\frac{M}{2} - 1\right)\right)$$

$$k_y = \frac{1}{N \cdot \Delta y}\left(\frac{-N}{2}, \cdots, 0 \cdots, \left(\frac{N}{2} - 1\right)\right)$$

When *M* or *N* is odd,

$$k_x = \frac{1}{M \cdot \Delta x}\left(\frac{-M-1}{2}, \cdots, 0 \cdots, \left(\frac{M-1}{2}\right)\right)$$

$$k_y = \frac{1}{N \cdot \Delta y}\left(\frac{-N-1}{2}, \cdots, 0 \cdots, \left(\frac{N-1}{2}\right)\right)$$

(11): In eq. (9), normal gravity is a function with respective to latitude.
**Response:** You are right that normal gravity is a function with respective to latitude. This study used the mean value of normal gravity. Thank you for this comment.

Line 159: How about north and east components of DOV?
**Response:** Yes, this analysis was performed for each of fused north and fused east components of DOV. When revising the manuscript, we will rephrase the wording of the sentence to make it clearer to understand.

Line 165: How to construct the eq.?
**Response:** The construction of Eq. 15 has been explained in lines 166 to 169. The left-hand-side of the equation is a vector of values of the fused $\xi$ component. The design matrix on the right-hand-side is made up of five column vectors (signifying five satellites), each vector contains values of $\xi$ component from one satellite. The unknown parameters (i.e., $a_1$, $a_2$, $a_3$, $a_4$, $a_5$) are solved through least squares to obtain the contribution of each satellite in resolving $\xi$.

Line 173: Jason GM?
**Response:** Yes, the two Jason GMs used in this study.

Line 178: What are both weighting approaches?
**Response:** Here, 'both weighting approaches' refers to the regional and global weight assignments. We will rephrase the sentence.

Line 185: SIO can provide DOV model. But DTU not provide directly DOV model.
**Response:** You are right. We stated this point clearly in lines 187 to 189. We will remove the words "and DTU (https://ftp.space.dtu.dk/pub/)".

(17): δg is generally for gravity disturbance, not gravity anomaly.
**Response:** Thank you for this comment. We will change it accordingly.

Line 130: The Laplacian equation only holds true outside the earth.
**Response:** We agree with you that Laplacian equation only holds true outside the earth. However, this same equation forms the theoretical basis on which the spectral relationships between $(\xi, \eta)$ and $\Delta g$, as well as $(\xi, \eta)$ and $T_{zz}$ were established in the study of Smith and Sandwell (1997). The derivation of these relationships can be seen in Appendix A of their paper. So, similarly in this paper, the Laplacian equation provides a theoretical check on the accuracy of our results.

Sandwell, D. T. and Smith, W. H. F.: Marine gravity anomaly from Geosat and ERS 1 satellite altimetry, J. Geophys. Res., 102, 10039–10054, https://doi.org/10.1029/96JB03223, 1997.

Line 235: The number of decimal separator may be more.
**Response:** We will increase the number of decimal places in the next version of the manuscript. Thank you for this comment.

Line 255: How to distinguish wave lengths?
**Response:** Looking at Figure 9, the coherency curve of the inverted Tzz intersects the 0.5 coherency value at wavelengths of 20 and 345 km.

Figure 1: Here should be GMs.
**Response:** Thank you for this comment. We will change it accordingly.

Figure 3: The resolution of the figure is low.
**Response:** We agree that the resolution of the figure is low. Honestly speaking, the resolution of the original figure is still very high. So, we believe this was caused by the image-to-pdf conversion during the submission process. We will definitely provide the original high-resolution figure for publication. Thank you for pointing this out to us.

Figure 5: The resolution of the figure is low.
**Response:** Again, the explanation to the previous comment applies here too. The original resolution of this figure is also very high. Both figures 3 and 5 were created at 300 dpi.

---

## Author Comment (AC1)

Dear Editor and Reviewers,

Many thanks for your constructive comments which have helped us to improve the manuscript. The replies to the comments are as follows:

Reviewer #1:

In my opinion, this manuscript does not meet the standard of enabling the user to evaluate whether the data product CUGB2023GRAD will be useful, whether it is an improvement on existing products, or even whether or how the authors have taken steps to avoid circular reasoning. Critical issues in data editing, filtering, and computation are not adequately described. I also believe that the input data sets may not be the best ones for this kind of analysis.

**Response:** Many thanks for pointing out this weakness in our work. Since the day we first saw your comments on this study, we have been working on implementing the two-pass retracking technique in order to use the 10, 20 or 40 Hz altimetry datasets like you mentioned. However, we must admit that it is not an easy technique to implement. Honestly speaking, we sought help from an expert who informed us that it is "not easy to retrack the datasets in a short period within the revision request". Therefore, considering this time constraint, we have instead decided to use SIO's vertical deflection components as the inputs. Since they were inverted from high-frequency altimetry datasets through the two-pass retracking, we believe this will help to improve the quality of the resultant gravity gradient tensor. Please note that this will change the content of Section 2.

The work described seems to rely heavily on the previous works of the Danish group led by Ole Andersen and the Scripps group led by David Sandwell. While a few papers from these groups are cited, often the most relevant ones are not cited. It would be helpful to have reviews from Drs. Andersen and Sandwell.

**Response:** We are grateful for pointing us to the relevant papers. However, although previous works of Ole Andersen and David Sandwell are cited in this study, apart from the use of DTUUH22MDT in the preprocessing, any major reliance on previous works of Ole Andersen and David Sandwell is seen in the comparison parts of the Results and Analysis section. The derivation of gravity gradient tensor does not heavily rely on their previous works. We are now about to rely on SIO since we intend to use their vertical deflection components as the new inputs when revising the manuscript. Correspondingly, more papers on products of the SIO group will be certainly cited. Once again, many thanks for this comment.

The strong coherence of the data with the GEBCO bathymetry (manuscript Figure 9 and its discussion) should not be taken as a measure of the quality or value of CUGB2023GRAD, because depths predicted from satellite altimetry are in the GEBCO product. In order that the reasoning not be circular here, one would need to demonstrate that the coherence had been computed from a portion of the GEBCO product that contained almost entirely in situ measured depths, and not depths estimated from altimetry. The GEBCO Source ID grid could be a help here.

**Response:** We agree with you on this comment. We have therefore decided to use multibeam shipborne measurements from NCEI's Autogrid web tool instead of the GEBCO grid. Thank you very much.

As the manuscript makes clear, all six gradient tensor elements arise from differentiation of one scalar quantity (the disturbing potential), and so these elements are six different views or characterizations of one set of information. The manuscript does not demonstrate the utility of having all these different views of the same information.

**Response:** Thank you for this comment. We admit that the current version of the manuscript does not discuss the significance of having all six tensor components. In Wan et al. (2023), we analyzed the bathymetric capabilities of each tensor component. Interestingly, the commonly used $T_{zz}$ does not

yield the most accurate bathymetry; but rather *Txz* and *Tyz* are the most bathymetric-suitable gravity gradients. We will include such discussion when revising the manuscript.

Wan, X., Annan, R. F., and Ziggah, Y. Y.: Altimetry-Derived Gravity Gradients Using Spectral Method and Their Performance in Bathymetry Inversion Using Back-Propagation Neural Network, JGR Solid Earth, 128, https://doi.org/10.1029/2022JB025785, 2023.

The manuscript presents the trace (sum of the diagonal elements) of the gradient tensor and suggests that the quantitative value of this sum is an evaluation of the product. But since the equations used all derive from the assumption that the gravity field obeys Laplace's equation, the trace ought to be zero by definition. The manuscript does not present any way for the reader to understand quantitatively the significance of a non-zero trace: how does it compare to the noise in the data, noise in the model components, limitations on the resolution of each quantity, etc.?

**Response:** Thank you for this comment and the related suggestions. We will analyze the non-zero trace values in relation to the recommendations you have made.

The manuscript has too many equations presenting the general theory, as this could have been summarized with a citation to a standard textbook. Some of these equations are given in spherical coordinates and some in Cartesian coordinates, and it is not clear which coordinates are used for the calculations done to produce CUGB2023GRAD. What the manuscript needs to do is to explain how the calculations in the 2 degree by 2 degree patches of Earth surface area were carried out. I presume they used Fourier transforms on Cartesian coordinates after a remove-restore procedure.

**Response:** Thank you for this comment. We will summarize some of the equations, and rather refer readers to their respective cited papers. You are absolutely right that we used Fourier transforms after a remove-restore procedure, but not on Cartesian coordinates. FFT computation is conducted in terms of theta and lambda, but not x and y. In terms of theta and lambda, the spatial intervals are same on the whole Earth. The equations involving spherical coordinates were simply used to explain how to obtain the signals $\Delta T_{\varphi}$ and $\Delta T_{\lambda}$ . However, note that the Fourier transforms were done in much bigger patches of 20⁰×20⁰ with 1⁰ overlaps in longitude and latitude. They are finally merged to obtain a global model using the *grdblend* module of GMT.

Concerning the 2⁰×2⁰ patches, they were only used for assigning regional weights to each satellite in order to merge the five sources of vertical deflection components. So, instead of a single weight for a satellite's vertical deflection component in a global setting (of say 160⁰×360⁰), we assigned local weights in each 2⁰×2⁰ patch. Therefore, the vertical deflection components were merged locally in each 2⁰×2⁰ patch. They are finally combined to obtain a global vertical deflection component.

Kindly note that Sections 3.1 and 4.1 will be no longer be needed in the revised manuscript since we will now be using SIO's vertical deflection components.

In my opinion 10.1016/j.asr.2019.09.011 does a very good job of demonstrating quantitatively the contribution of each satellite altimeter mission to the overall marine gravity field model, treating the east-west and north-south components separately and treating each as functions of latitude, and showing what weight should be given to each. The present manuscript does not do this very well. Perhaps the present study's analysis of what weight to give each satellite mission in deriving each of the tensor quantities in each of the 2x2 squares might furnish some interesting information, but that information is not presented here.

**Response:** Thank you for suggesting to us another approach of analyzing the contributions of each satellite altimeter mission. Also, we appreciate your suggestion about weighting each satellite mission when deriving each of the tensor quantities in each of the 2x2 squares. However, we cannot

implement either suggestion on the already merged vertical deflections from SIO. We will definitely experiment with these ideas in our future studies.

The filtering of the data is an important detail, but the equation describing the filter (Equation 1) is wrong: if tau is the filter width parameter then tau-squared should appear somewhere in the argument to the exponential in Equation 1. Another minor point: I assume that the data to be filtered are very closely spaced, and in that case computing the filter using Equation 2 followed by the inverse cosine will be quite inaccurate.

**Response:** Many thanks for pointing this out to us.

The input data used are available only at the "1 Hz" nominal sampling rate, for many of the altimeters included in this study. As many papers by Sandwell and his colleagues for over 30 years have shown, "1 Hz" data are down-sampled from boxcar averages of the original data, which have nominal sampling rates of 10, 20 or 40 Hz, depending on the satellite; the boxcar averaging has bad side-lobes; and the 1-Hz downsampling aliases sidelobe energy into long along-track wavelengths, spoiling the accuracy of the resulting along-track deflections of the vertical. For this reason, Sandwell and colleagues have taken great pains to design specialized filters and down sampling rates. Therefore I believe that the accuracy and utility of CUGB2023GRAD may be limited by the fact that it starts from "1 Hz" data. (The along-track filter design description in 10.1029/95JB01308 pre-dates the development of two-pass retracking [10.1093/gji/ggt469] and so the filters now used have different pass- and stop-band specification than what is described in 95JB01308.)

**Response:** We acknowledge that the main weakness of this version of the manuscript is the 1 Hz input data used. Like we have expressed in our preceding responses, we have therefore decided to replace the input vertical deflections with SIO's. We are very grateful for this constructive comment.

An important detail is the removal of the non-geoidal signals from the sea surface height. One of these, the Mean Dynamic Topography, is mentioned in Equation 3. But others (tides, transient dynamic signals, etc., as well as errors in radar path delays, sea state bias, etc.) are not mentioned. One wonders how this was done. It will have an important impact on the quality of CUGB2023GRAD.

**Response:** You are right that the removal of the non-geoidal signals from the sea surface height is an important step. Indeed, we performed these steps during the preprocessing stage. This included errors due to: sea state bias, solid earth tide, polar tide, ocean tide, wet troposphere, dry troposphere, ionosphere, dynamic atmospheric, etc. All of these corrections were computed according to the parameter thresholds stipulated in Along-track Level-2+ (L2P) SLA Product Handbook.

Equation (4) correctly shows that the north and east components of deflection are the corresponding partial derivatives of the geoid height anomaly, but in the Introduction these deflection components are incorrectly described as derivatives of the disturbing potential. The correct relationship requires relating the geoid height to the disturbing potential, such as via Bruns formula.

**Response:** Thank you for pointing this out to us. We will change it accordingly.

It is not correct to say there has been little prior work on the vertical gravity gradient; in addition to 10.1126/science.1258213 , 10.1029/2020JB020017 should also be cited.

**Response:** We think you misunderstood us. We said most previous works about gravity gradients are themed on the vertical gravity gradient even though there are five other gravity gradients which also are worth researching on. So, we did not say there has been little prior work on the vertical gravity gradient. Rather, what we intended to say is that the full gravity gradient tensor should be studied instead of only focusing on vertical gravity gradient. Again, thank you for pointing us to another paper that should be cited.

The Generic Mapping Tools should be cited for the program *surface* at line 111; which citation depends on which version was used: https://www.generic-mapping-tools.org/cite/

**Response:** Alright, we will add the citation as you have recommended. Thank you very much.

Reviewer #2:

All over the paper, "deflection of the vertical" should be "deflections of the vertical", including both north-south and west-east components.

**Response:** Thanks for the comment. We will make in the changes when revising the manuscript.

Comment: Line 9, in the abstract, "They are derived from double differentiation of the geoid (or disturbing potential)". Its not clear. Gravity gradients are the second second derivative of disturbing potential. The geoid should be connected with disturbing potential by Bruns formula

**Response:** You are right, our attention has been drawn to it. We will change the wording to: "They are derived from double differentiation of the disturbing potential".

Comment: Line 37, altimetric gravity anomaly is widely used to predict bathymetry, too few references are given. Many paper from Smith, Sandwell and Anderson should be referenced..

**Response:** Many thanks for this comment. Papers from the aforementioned researchers will be cited in the revised version of the manuscript.

Comment: Line51-52, "Deflection of the vertical and gravity anomaly are its first derivatives in the horizontal and vertical directions, respectively. ", Again, you should make it clear, geoid and disturbing potential are different.

**Response:** Sure, you are right. We will change it to disturbing potential instead of geoid.

Comment: Line 130,the first formula in eq.(7), there should be a "-"

**Response:** Thanks very much. We will make the correction.

Comment: In order to validate the gravity gradients results, the coherence between CUGB2023GRAD and GEBCO_2021 were computed. In order to avoid circular validation problems, I recommend adopting multi-beam bathymetry grid from JAMSTEC or NCEI to replace GEBCO_2021 model.

**Response:** This recommendation is very helpful. We will get multibeam depths from NCEI's Autogrid web tool to replace GEBCO_2021. Thank you very much.

Reviewer #3:

The paper does not explain the strength of the method applied nor the potential benefit of their product for use which I think is very important. On the one hand, the methodology needs to be expanded in general. But on the other hand, there are various equations which I think is very lengthy. Some adjustment is needed therefore in balancing the content. I am not sure if an Appendix can be added.

**Response:** Generally, few investigations discussed full tensors gravity gradients inversion using altimetry data, and the main products provided by altimetry satellites are gravity anomaly, and vertical deflections. This study adopted Fourier transform approach to derive full tensors gravity gradients. The strength of this study is that it presents an easier and faster approach to compute all six components of the gravity gradient tensor, especially from altimetry-derived deflection of the vertical.

On the potential application of the product, it can be used to invert bathymetry to fill gaps between ship tracks. Kindly see Wan et al. (2023), in which we analyzed the bathymetric capabilities of each tensor component. Interestingly, the commonly used *Tzz* does not yield the most accurate bathymetry; but rather *Txz* and *Tyz* are the most bathymetric-suitable gravity gradients. It can also be used for identifying seamounts; such as in Kim and Wessel (2015) and Wessel et al. (2022).

Kim, S.-S. and Wessel, P.: Finding seamounts with altimetry-derived gravity data, in: OCEANS 2015 - MTS/IEEE Washington, OCEANS 2015 - MTS/IEEE Washington, Washington, DC, 1–6, https://doi.org/10.23919/OCEANS.2015.7401883, 2015.

Wan, X., Annan, R. F., and Ziggah, Y. Y.: Altimetry-Derived Gravity Gradients Using Spectral Method and Their Performance in Bathymetry Inversion Using Back-Propagation Neural Network, JGR Solid Earth, 128, https://doi.org/10.1029/2022JB025785, 2023.

Wessel, P., Watts, A. B., Kim, S.-S., and Sandwell, D. T.: Models for the evolution of seamounts, Geophysical Journal International, 231, 1898–1916, https://doi.org/10.1093/gji/ggac285, 2022.

I can imagine readers being interested in learning more about the benefits and applications of the outcome of this work. The authors could be more convincing and provide some details also w.r.t. the literature. This is not provided in the current version. I also think that some of the figures can be explained better.

**Response:** You are absolutely right. Like we have answered in the previous response, we will include a section which entails the product's potential benefit in the revised version of the manuscript. Also, we will work on rephrasing the explanations of the figures to make them more understandable. We would be more grateful if you could pinpoint to us the figures you think require additional explanations.

The use of EGM2008 needs to be justified in general. For marine related gravity field, I can imagine GOCE data also should be included in the GGM used. I wonder whether GEBCO grid is the best option to compare with. Would there be any case one compares the outcome of this work for instance w.r.t shipborne measurements directly, maybe along track measurements of bathymetry?

**Response:** You are right that the GGM used should contain GOCE data. Our main reason for using EGM2008 is that, it is the most widely used GGM for studies involving marine gravity field (Sandwell et al., 2019; Zhang et al., 2017; Zhu et al., 2020, 2019; Andersen and Knudsen, 2019). Concerning the use of GEBCO grid, we will replace it with multibeam shipborne measurements from NCEI's Autogrid web tool. Many thanks for this comment.

Andersen, O. B. and Knudsen, P.: The DTU17 Global Marine Gravity Field: First Validation Results, in: Fiducial Reference Measurements for Altimetry, vol. 150, edited by: Mertikas, S. P. and Pail, R., Springer, Cham, 83–87, 2019.

Sandwell, D. T., Harper, H., Tozer, B., and Smith, W. H. F.: Gravity field recovery from geodetic altimeter missions, Advances in Space Research, S0273117719306593, https://doi.org/10.1016/j.asr.2019.09.011, 2019.

Zhang, S., Sandwell, D. T., Jin, T., and Li, D.: Inversion of marine gravity anomalies over southeastern China seas from multi-satellite altimeter vertical deflections, Journal of Applied Geophysics, 10, https://doi.org/10.1016/j.jappgeo.2016.12.014, 2017.

Zhu, C., Guo, J., Hwang, C., Gao, J., Yuan, J., and Liu, X.: How HY-2A/GM altimeter performs in marine gravity derivation: assessment in the South China Sea, Geophysical Journal International, 219, 1056–1064, https://doi.org/10.1093/gji/ggz330, 2019.

Zhu, C., Guo, J., Gao, J., Liu, X., Hwang, C., Yu, S., Yuan, J., Ji, B., and Guan, B.: Marine gravity determined from multi-satellite GM/ERM altimeter data over the South China Sea: SCSGA V1.0, J Geod, 94, 50, https://doi.org/10.1007/s00190-020-01378-4, 2020.

Line 101: Please use the ESSD reference for the ICGEM service.

**Response:** Thank you for this comment. We will make the correction when revising the manuscript.

Line 111: Please explain why the weights are calculated based on EGM2008? What would be an alternative to this?

**Response:** Like we have answered in our earlier response, the weights are calculated based on EGM2008 because, EGM2008 is the most commonly used GGM in studies involving marine gravity inversion. An alternative would be XGM2019e or EIGEN-6C4.

Line 133: Preference for using EGM2008 needs to be justified.

**Response:** We have replied to this comment in the preceding responses.

Last sentence on Page 6 is confusing.

**Response:** Alright, we will change the wording of that sentence. What we meant was that:

From Eq. (7),

$$\xi = \frac{1}{\gamma r} \cdot \frac{\partial T}{\partial \varphi} \qquad \frac{\partial T}{\partial \varphi} = T_\varphi = \xi \cdot \gamma r$$
$$\eta = -\frac{1}{\gamma r \sin \varphi} \cdot \frac{\partial T}{\partial \lambda} \quad \Rightarrow \quad \frac{\partial T}{\partial \lambda} = T_\lambda = -\eta \cdot \gamma r \sin \varphi$$

If we replace $\xi$ and $\eta$ by their residual versions $\Delta\xi$ and $\Delta\eta$, then we will get:

$$\Delta T_\varphi = \Delta\xi \cdot \gamma r$$
$$\Delta T_\lambda = -\Delta\eta \cdot \gamma r \sin \varphi$$

Line 171: Why somewhat? Was it unexpected?

**Response:** We expected similar ranking as theirs. We used 'somewhat' because we did not have exactly the same conclusion as theirs. 'Somewhat' in this context means 'slightly'. In this study, Saral/AltiKa ranked ahead of Cryosat-2; whereas in Zhu et al. (2020), Cryosat-2 outranked Saral/AltiKa. The remainder of the ranking is same as theirs.

Again, some sections are unnecessarily lengthy and some sections are not explained as much needed.

**Response:** Thank you for this comment. We will work on that when revising the manuscript.

Line 193: No parenthesis for deflections of vertical

**Response:** Alright, we will remove the parenthesis.

Line 301: GEBCO_2021 or 2022? I believe the two are used in the paper. Please explain why? Would there be any in-situ data available for this purpose? Would it be more reliable to do comparisons w.r.t shipborne or other in-situ measurements?

**Response:** No, we used GEBCO_2022. If there is GEBCO_2021 in the manuscript, then it is a typo. Again, per your earlier advice, we will use shipborne depth measurements instead of the GEBCO grid in the next version of the manuscript.

---

## Author Response (AR1)

Dear Editor and Reviewers,
Many thanks for your constructive comments which have helped us to improve the manuscript. We have modified the manuscript according to the comments carefully. The revised contents are highlighted in red in the manuscript. The replies to the comments are as follows:

Reviewer #1:

In my opinion, this manuscript does not meet the standard of enabling the user to evaluate whether the data product CUGB2023GRAD will be useful, whether it is an improvement on existing products, or even whether or how the authors have taken steps to avoid circular reasoning. Critical issues in data editing, filtering, and computation are not adequately described. I also believe that the input data sets may not be the best ones for this kind of analysis.

**Response:** Many thanks for pointing out this weakness in our work. Since the day we first saw your comments on this study, we have been working on implementing the two-pass retracking technique in order to use the 10, 20 or 40 Hz altimetry datasets like you mentioned. However, we must admit that it is not an easy technique to implement. Honestly speaking, we sought help from an expert who informed us that it is "not easy to retrack the datasets in a short period within the revision request". Therefore, considering this time constraint, we have instead resorted to the average of SIO's vertical deflection components and vertical deflection components we inverted from DTU21GRA. Since these were inverted from high-frequency altimetry datasets through the two-pass retracking, we believe this will help to improve the quality of the resultant gravity gradient tensor. Please note that this will change the content of Section 2.

The work described seems to rely heavily on the previous works of the Danish group led by Ole Andersen and the Scripps group led by David Sandwell. While a few papers from these groups are cited, often the most relevant ones are not cited. It would be helpful to have reviews from Drs. Andersen and Sandwell.

**Response:** We are grateful for pointing us to the relevant papers. However, although previous works of Ole Andersen and David Sandwell are cited in this study, apart from the use of DTUUH22MDT in the preprocessing, any major reliance on previous works of Ole Andersen and David Sandwell was seen in the comparison parts of the Results and Analysis section. The derivation of gravity gradient tensor does not heavily rely on their previous works. Correspondingly, more papers on products of the SIO and DTU groups have been cited in this version of the manuscript. Once again, many thanks for this comment.

The strong coherence of the data with the GEBCO bathymetry (manuscript Figure 9 and its discussion) should not be taken as a measure of the quality or value of CUGB2023GRAD, because depths predicted from satellite altimetry are in the GEBCO product. In order that the reasoning not be circular here, one would need to demonstrate that the coherence had been computed from a portion of the GEBCO product that contained almost entirely in situ measured depths, and not depths estimated from altimetry. The GEBCO Source ID grid could be a help here.

**Response:** We agree with you on this comment. We have therefore decided to use multibeam shipborne measurements from NCEI's Autogrid web tool instead of the GEBCO grid. Thank you very much.

As the manuscript makes clear, all six gradient tensor elements arise from differentiation of one scalar quantity (the disturbing potential), and so these elements are six different views or characterizations of one set of information. The manuscript does not demonstrate the utility of having all these different views of the same information.

**Response:** Thank you for this comment. We admit that the previous version of the manuscript does not discuss the significance of having all six tensor components. We have now included in Section

4.2, a discussion on the utility of having all six tensor components. The results agree with findings in our previous work in Wan et al. (2023). The frequently used *Tzz* is not the most influential gravity gradient for bathymetric studies. Rather, the rarely used non-diagonal gradients are the three most dominant signals for bathymetric prediction.

Wan, X., Annan, R. F., and Ziggah, Y. Y.: Altimetry-Derived Gravity Gradients Using Spectral Method and Their Performance in Bathymetry Inversion Using Back-Propagation Neural Network, JGR Solid Earth, 128, https://doi.org/10.1029/2022JB025785, 2023.

The manuscript presents the trace (sum of the diagonal elements) of the gradient tensor and suggests that the quantitative value of this sum is an evaluation of the product. But since the equations used all derive from the assumption that the gravity field obeys Laplace's equation, the trace ought to be zero by definition. The manuscript does not present any way for the reader to understand quantitatively the significance of a non-zero trace: how does it compare to the noise in the data, noise in the model components, limitations on the resolution of each quantity, etc.?

**Response:** Thank you for this comment and the related suggestions. After using the average of vertical deflections from SIO and DTU, the sum of the diagonal elements now shows zero across the globe.

The manuscript has too many equations presenting the general theory, as this could have been summarized with a citation to a standard textbook. Some of these equations are given in spherical coordinates and some in Cartesian coordinates, and it is not clear which coordinates are used for the calculations done to produce CUGB2023GRAD. What the manuscript needs to do is to explain how the calculations in the 2 degree by 2 degree patches of Earth surface area were carried out. I presume they used Fourier transforms on Cartesian coordinates after a remove-restore procedure.

**Response:** Thank you for this comment. We have now summarized most of the equations. Due to the current use of vertical deflections directly from SIO and DTU, equations related to the Section 3.1 in the previous version of the manuscript have now been removed. You are absolutely right that we used Fourier transforms after a remove-restore procedure, but not on Cartesian coordinates. FFT computation is conducted in terms of theta and lambda, but not x and y. In terms of theta and lambda, the spatial intervals are same on the whole Earth. The equations involving spherical coordinates were simply used to explain how to obtain the signals $\Delta T_{\varphi}$ and $\Delta T_{\lambda}$. However, note that the Fourier transforms were done in much bigger patches of 20⁰×20⁰ with 1⁰ overlaps in longitude and latitude. They are finally merged to obtain a global model using the *grdblend* module of GMT.

Concerning the 2⁰×2⁰ patches, they were only used for assigning regional weights to each satellite in order to merge the five sources of vertical deflection components. So, instead of a single weight for a satellite's vertical deflection component in a global setting (of say 160⁰×360⁰), we assigned local weights in each 2⁰×2⁰ patch. Therefore, the vertical deflection components were merged locally in each 2⁰×2⁰ patch. They are finally combined to obtain a global vertical deflection component.

Kindly note that Section 3.1 has been completely modified; and Section 4.1 now discusses the inverted gravity gradient tensor.

In my opinion 10.1016/j.asr.2019.09.011 does a very good job of demonstrating quantitatively the contribution of each satellite altimeter mission to the overall marine gravity field model, treating the east-west and north-south components separately and treating each as functions of latitude, and showing what weight should be given to each. The present manuscript does not do this very well. Perhaps the present study's analysis of what weight to give each satellite mission in deriving each of the tensor quantities in each of the 2x2 squares might furnish some interesting information, but that information is not presented here.

**Response:** Thank you for suggesting to us another approach of analyzing the contributions of each satellite altimeter mission. Also, we appreciate your suggestion about weighting each satellite mission when deriving each of the tensor quantities in each of the 2x2 squares. However, we cannot implement either suggestion on the already merged datasets from SIO and DTU. We will definitely experiment with these ideas in our future studies.

The filtering of the data is an important detail, but the equation describing the filter (Equation 1) is wrong: if tau is the filter width parameter then tau-squared should appear somewhere in the argument to the exponential in Equation 1. Another minor point: I assume that the data to be filtered are very closely spaced, and in that case computing the filter using Equation 2 followed by the inverse cosine will be quite inaccurate.

**Response:** Many thanks for pointing this out to us.

The input data used are available only at the "1 Hz" nominal sampling rate, for many of the altimeters included in this study. As many papers by Sandwell and his colleagues for over 30 years have shown, "1 Hz" data are down-sampled from boxcar averages of the original data, which have nominal sampling rates of 10, 20 or 40 Hz, depending on the satellite; the boxcar averaging has bad side-lobes; and the 1-Hz downsampling aliases sidelobe energy into long along-track wavelengths, spoiling the accuracy of the resulting along-track deflections of the vertical. For this reason, Sandwell and colleagues have taken great pains to design specialized filters and down sampling rates. Therefore I believe that the accuracy and utility of CUGB2023GRAD may be limited by the fact that it starts from "1 Hz" data. (The along-track filter design description in 10.1029/95JB01308 pre-dates the development of two-pass retracking [10.1093/gji/ggt469] and so the filters now used have different pass- and stop-band specification than what is described in 95JB01308.)

**Response:** We acknowledge that the main weakness of the previous version of the manuscript was the 1 Hz input data used. Like we have expressed in our preceding responses, we have now replaced the inputs with the average of SIO's and DTU21GRA-derived vertical deflections. We are very grateful for this constructive comment.

An important detail is the removal of the non-geoidal signals from the sea surface height. One of these, the Mean Dynamic Topography, is mentioned in Equation 3. But others (tides, transient dynamic signals, etc., as well as errors in radar path delays, sea state bias, etc.) are not mentioned. One wonders how this was done. It will have an important impact on the quality of CUGB2023GRAD.

**Response:** You are right that the removal of the non-geoidal signals from the sea surface height is an important step. Indeed, we performed these steps during the preprocessing stage. This included errors due to: sea state bias, solid earth tide, polar tide, ocean tide, wet troposphere, dry troposphere, ionosphere, dynamic atmospheric, etc. All of these corrections were computed according to the parameter thresholds stipulated in Along-track Level-2+ (L2P) SLA Product Handbook. Once again, because of the current use of already developed vertical deflections from SIO and DTU, this step is no longer needed in the current version of the manuscript.

Equation (4) correctly shows that the north and east components of deflection are the corresponding partial derivatives of the geoid height anomaly, but in the Introduction these deflection components are incorrectly described as derivatives of the disturbing potential. The correct relationship requires relating the geoid height to the disturbing potential, such as via Bruns formula.

**Response:** Thank you for pointing this out to us. We have changed it accordingly.

It is not correct to say there has been little prior work on the vertical gravity gradient; in addition to 10.1126/science.1258213 , 10.1029/2020JB020017 should also be cited.

**Response:** We think you misunderstood us. We said most previous works about gravity gradients are themed on the vertical gravity gradient even though there are five other gravity gradients which also are worth researching on. So, we did not say there has been little prior work on the vertical gravity gradient. Rather, what we intended to say is that the full gravity gradient tensor should be studied instead of only focusing on vertical gravity gradient. We have modified the wording of that statement to avoid ambiguity. Again, thank you for pointing us to another paper that should be cited.

The Generic Mapping Tools should be cited for the program *surface* at line 111; which citation depends on which version was used: https://www.generic-mapping-tools.org/cite/

**Response:** The current version of the manuscript does not mention *surface*; however, we used it during the analyses. So, we have cited it as part of the Acknowledgements section. Thank you very much.

Reviewer #2:

All over the paper, "deflection of the vertical" should be "deflections of the vertical", including both north-south and west-east components.

**Response:** Thanks for the comment. We have made in the changes.

Comment: Line 9, in the abstract, "They are derived from double differentiation of the geoid (or disturbing potential)". Its not clear. Gravity gradients are the second second derivative of disturbing potential. The geoid should be connected with disturbing potential by Bruns formula

**Response:** You are right, our attention has been drawn to it. We have changed the wording to: "They are derived from double differentiation of the disturbing potential".

Comment: Line 37, altimetric gravity anomaly is widely used to predict bathymetry, too few references are given. Many paper from Smith, Sandwell and Anderson should be referenced..

**Response:** Many thanks for this comment. Additional papers from the aforementioned researchers now have been cited in this revised version of the manuscript.

Comment: Line51-52, "Deflection of the vertical and gravity anomaly are its first derivatives in the horizontal and vertical directions, respectively. ", Again, you should make it clear, geoid and disturbing potential are different.

**Response:** Sure, you are right. We have changed it to disturbing potential instead of geoid.

Comment: Line 130,the first formula in eq.(7), there should be a "-"

**Response:** Thanks very much. We have made the correction.

Comment: In order to validate the gravity gradients results, the coherence between CUGB2023GRAD and GEBCO_2021 were computed. In order to avoid circular validation problems, I recommend adopting multi-beam bathymetry grid from JAMSTEC or NCEI to replace GEBCO_2021 model.

**Response:** This recommendation is very helpful. We have replaced it with multibeam depths of the Tonga Trench obtained from NCEI's Autogrid web tool. Thank you very much.

Reviewer #3:

The paper does not explain the strength of the method applied nor the potential benefit of their product for use which I think is very important. On the one hand, the methodology needs to be expanded in general. But on the other hand, there are various equations which I think is very lengthy. Some adjustment is needed therefore in balancing the content. I am not sure if an Appendix can be added.

**Response:** Generally, few investigations have discussed full tensors gravity gradients inversion using altimetry data, and the main products provided by altimetry satellites are gravity anomaly, and vertical deflections. This study adopted Fourier transform approach to derive full tensors gravity gradients. The strength of this study is that it presents an easier and faster approach to compute all six components of the gravity gradient tensor, especially from altimetry-derived deflection of the vertical.

On the potential application of the product, it can be used to invert bathymetry to fill gaps between ship tracks. We now included Section 4.2, which discusses the utility of having all six tensor components. We assessed the bathymetric influence of each tensor component through a deep learning bathymetry inversion approach we developed in Annan and Wan (2022). The results agree with findings in our previous work in Wan et al. (2023). Interestingly, the commonly used $Tzz$ does not yield the most accurate bathymetry; but rather $Txy$, $Txz$ and $Tyz$ are the most bathymetric-suitable gravity gradients. The product can also be used for identifying seamounts; such as in Kim and Wessel (2015) and Wessel et al. (2022).

Annan, R. F. and Wan, X.: Recovering Bathymetry of the Gulf of Guinea Using Altimetry-Derived Gravity Field Products Combined via Convolutional Neural Network, Surv Geophys, https://doi.org/10.1007/s10712-022-09720-5, 2022.

Kim, S.-S. and Wessel, P.: Finding seamounts with altimetry-derived gravity data, in: OCEANS 2015 - MTS/IEEE Washington, OCEANS 2015 - MTS/IEEE Washington, Washington, DC, 1–6, https://doi.org/10.23919/OCEANS.2015.7401883, 2015.

Wan, X., Annan, R. F., and Ziggah, Y. Y.: Altimetry-Derived Gravity Gradients Using Spectral Method and Their Performance in Bathymetry Inversion Using Back-Propagation Neural Network, JGR Solid Earth, 128, https://doi.org/10.1029/2022JB025785, 2023.

Wessel, P., Watts, A. B., Kim, S.-S., and Sandwell, D. T.: Models for the evolution of seamounts, Geophysical Journal International, 231, 1898–1916, https://doi.org/10.1093/gji/ggac285, 2022.

I can imagine readers being interested in learning more about the benefits and applications of the outcome of this work. The authors could be more convincing and provide some details also w.r.t. the literature. This is not provided in the current version. I also think that some of the figures can be explained better.

**Response:** You are absolutely right. Like we have answered in the previous response, we have now included Section 4.2 which discusses the product's potential benefit in this revised version of the manuscript. Also, we have rephrased the explanations of the figures to make them more understandable. We would be more grateful if you could pinpoint to us figures you think require additional explanations.

The use of EGM2008 needs to be justified in general. For marine related gravity field, I can imagine GOCE data also should be included in the GGM used. I wonder whether GEBCO grid is the best option to compare with. Would there be any case one compares the outcome of this work for instance w.r.t shipborne measurements directly, maybe along track measurements of bathymetry?

**Response:** You are right that the GGM used should contain GOCE data. Our main reason for using EGM2008 is that, it is the most widely used GGM for studies involving marine gravity field (Sandwell et al., 2019; Zhang et al., 2017; Zhu et al., 2020, 2019; Andersen and Knudsen, 2019). Concerning the use of GEBCO grid, we have replaced it with multibeam shipborne measurements of the Tonga Trench from NCEI's Autogrid web tool. Many thanks for this comment.

Andersen, O. B. and Knudsen, P.: The DTU17 Global Marine Gravity Field: First Validation Results, in: Fiducial Reference Measurements for Altimetry, vol. 150, edited by: Mertikas, S. P. and Pail, R., Springer, Cham, 83–87, 2019.

Sandwell, D. T., Harper, H., Tozer, B., and Smith, W. H. F.: Gravity field recovery from geodetic altimeter missions, Advances in Space Research, S0273117719306593, https://doi.org/10.1016/j.asr.2019.09.011, 2019.

Zhang, S., Sandwell, D. T., Jin, T., and Li, D.: Inversion of marine gravity anomalies over southeastern China seas from multi-satellite altimeter vertical deflections, Journal of Applied Geophysics, 10, https://doi.org/10.1016/j.jappgeo.2016.12.014, 2017.

Zhu, C., Guo, J., Hwang, C., Gao, J., Yuan, J., and Liu, X.: How HY-2A/GM altimeter performs in marine gravity derivation: assessment in the South China Sea, Geophysical Journal International, 219, 1056–1064, https://doi.org/10.1093/gji/ggz330, 2019.

Zhu, C., Guo, J., Gao, J., Liu, X., Hwang, C., Yu, S., Yuan, J., Ji, B., and Guan, B.: Marine gravity determined from multi-satellite GM/ERM altimeter data over the South China Sea: SCSGA V1.0, J Geod, 94, 50, https://doi.org/10.1007/s00190-020-01378-4, 2020.

Line 101: Please use the ESSD reference for the ICGEM service.

**Response:** Thank you for this comment. We made the correction in the current version of the manuscript.

Line 111: Please explain why the weights are calculated based on EGM2008? What would be an alternative to this?

**Response:** Like we have answered in our earlier response, this is not an issue because EGM2008 is the most commonly used GGM in studies involving marine gravity inversion. An alternative would be XGM2019e or EIGEN-6C4.

Line 133: Preference for using EGM2008 needs to be justified.

**Response:** We have replied to this comment in the preceding responses.

Last sentence on Page 6 is confusing.

**Response:** Alright, we have changed the wording of that sentence. What we meant was that:

From Eq. (7),

$$\xi = \frac{1}{\gamma r} \cdot \frac{\partial T}{\partial \varphi} \qquad \frac{\partial T}{\partial \varphi} = T_\varphi = \xi \cdot \gamma r$$
$$\Rightarrow$$
$$\eta = -\frac{1}{\gamma r \sin \varphi} \cdot \frac{\partial T}{\partial \lambda} \qquad \frac{\partial T}{\partial \lambda} = T_\lambda = -\eta \cdot \gamma r \sin \varphi$$

If we replace $\xi$ and $\eta$ by their residual versions $\Delta \xi$ and $\Delta \eta$, then we will get:

$$\Delta T_\varphi = \Delta \xi \cdot \gamma r$$
$$\Delta T_\lambda = -\Delta \eta \cdot \gamma r \sin \varphi$$

Please note that in the current version of the manuscript, this is now Eq. (3).

Line 171: Why somewhat? Was it unexpected?

**Response:** We expected similar ranking as theirs. We used 'somewhat' because we did not have exactly the same conclusion as theirs. 'Somewhat' in this context means 'slightly'. In the previous version of the manuscript, Saral/AltiKa ranked ahead of Cryosat-2; whereas in Zhu et al. (2020), Cryosat-2 outranked Saral/AltiKa. The remainder of the ranking is same as theirs.

The current version of the manuscript no longer contains this information due to the direct use of vertical deflections from Scripps Institution of Oceanography (Sandwell et al., 2019), and vertical deflections we inverted from DTU21GRA – a gravity anomaly model developed by Technical University of Denmark (Andersen et al., 2023).

Andersen, O. B., Rose, S. K., Abulaitijiang, A., Zhang, S., and Fleury, S.: The DTU21 global mean sea surface and first evaluation, Earth Syst. Sci. Data, 15, 4065–4075, https://doi.org/10.5194/essd-15-4065-2023, 2023.

Sandwell, D. T., Harper, H., Tozer, B., and Smith, W. H. F.: Gravity field recovery from geodetic altimeter missions, Advances in Space Research, S0273117719306593, https://doi.org/10.1016/j.asr.2019.09.011, 2019.

Again, some sections are unnecessarily lengthy and some sections are not explained as much needed.

**Response:** Thank you for this comment. We have revised the manuscript to make it more precise and understandable.

Line 193: No parenthesis for deflections of vertical

**Response:** Alright, we have removed the parenthesis.

Line 301: GEBCO_2021 or 2022? I believe the two are used in the paper. Please explain why? Would there be any in-situ data available for this purpose? Would it be more reliable to do comparisons w.r.t shipborne or other in-situ measurements?

**Response:** No, we used GEBCO_2022. If there is GEBCO_2021 in the manuscript, then it is a typo. Again, per your earlier advice, we have used multibeam depth measurements instead of the GEBCO grid in the current version of the manuscript.